BRIEF COMMUNICATION

# Dynamic forecasting of severe acute graft-versus-host disease after transplantation

Xueou Liu [1,3], Yigeng Cao[1,3], Ye Guo[1,3], Xiaowen Gong[1,3], Yahui Feng[1,3], Yao Wang [2,3], Mingyang Wang[1], Mengxuan Cui[2], Wenwen Guo[1], Luyang Zhang[1], Ningning Zhao[1], Xiaoqiang Song[1], Xuetong Zheng[1], Xia Chen[1], Qiujin Shen[1], Song Zhang[1], Zhen Song[1], Linfeng Li[2], Sizhou Feng[1], Mingzhe Han[1], Xiaofan Zhu [1✉], Erlie Jiang [1✉] and Junren Chen [1✉]

Forecasting of severe acute graft-versus-host disease (aGVHD) after transplantation is a challenging 'large $p$, small $n$' problem that suffers from nonuniform data sampling. We propose a dynamic probabilistic algorithm, daGOAT, that accommodates sampling heterogeneity, integrates multidimensional clinical data and continuously updates the daily risk score for severe aGVHD onset within a two-week moving window. In the studied cohorts, the cross-validated area under the receiver operator characteristic curve (AUROC) of daGOAT rose steadily after transplantation and peaked at ≥0.78 in both the adult and pediatric cohorts, outperforming the two-biomarker MAGIC score, three-biomarker Ann Arbor score, peri-transplantation features-based models and XGBoost. Simulation experiments indicated that the daGOAT algorithm is well suited for short time-series scenarios where the underlying process for event generation is smooth, multidimensional and where there are frequent and irregular data missing. daGOAT's broader utility was demonstrated by performance testing on a remotely different task, that is, prediction of imminent human postural change based on smartphone inertial sensor time-series data.

To this day, severe acute graft-versus-host disease (grade III–IV aGVHD) remains a leading cause of death after allogeneic hematopoietic stem cell transplantation (allo-HSCT)—a last-resort treatment for many blood diseases—with a transplant-related mortality rate as high as ~30% within 100 days[1].

Previous algorithms for forecasting severe aGVHD were usually based on peri-transplantation features (including recipient, donor and transplantation procedural parameters) or 'landmark' biomarker analysis (designating a specific time point, post-transplant, for plasma biomarker analysis). Area under the receiver operator characteristic curve (AUROC) scores of models using only peri-transplantation features were reported to be ~0.62, even when data from more than 20,000 patients were available[2,3]. For landmark analysis, it has been reported in at least some study cohorts that suppression of tumorigenicity 2 (ST2)—by far the most promising biomarker for predicting treatment response 'after' aGVHD onset[4,5]—had either no substantial association with aGVHD[6] or a low AUROC (0.56) for forecasting severe aGVHD[5] if measured at days 11 to 17 post-transplant (that is, 'before' aGVHD onset). One study of a Japanese cohort, however, did show association (AUROC 0.66) between grade II–IV aGVHD and ST2 measured on day 14[7]. A two-biomarker model using ST2 and regenerating islet-derived 3-alpha (Reg3α) measured on day 7 post-transplant—the MAGIC score—was shown to predict six-month non-relapse mortality (NRM; AUROC 0.68)[8]. Forecasting NRM, however, was not equivalent to forecasting severe aGVHD. Using data reported by Hartwell et al.[8] (their table S6), one could calculate that the proportion of NRM cases attributed to aGVHD was statistically indistinguishable between MAGIC score-stratified 'high-risk' and 'low-risk' groups (64% (41/64) versus 57% (47/83); $P = 0.458$, $\chi^2$ test). In contrast, if computed 'at' or 'after' aGVHD onset, multi-biomarker scores were shown to be efficacious in predicting treatment response and long-term outcome[8–10].

Few studies have used time-series of all available patient information for forecasting severe aGVHD. One recent study applied penalized logistic regression to vital signs (body temperature, heart rate and so on) that were consistently recorded within the first 10 days after HSCT and achieved an AUROC value of 0.66 for forecasting grade II–IV aGVHD[11]. Modeling in HSCT—unlike in more common medical situations—is challenged by the small sample size ($n$), high feature number ($p$) and nonuniform data sampling[12]. This study thus aims to utilize evidence from all available dynamic variables to forecast severe aGVHD better.

We compiled and curated the post-transplant multidimensional time-series data of patients treated with human leukocyte antigen (HLA)-mismatched allo-HSCT using stem cells derived from peripheral blood, bone marrow or both at the Institute of Hematology, Chinese Academy of Medical Sciences (IHCAMS) between 2012 and 2021 (hereafter referred to as the 'aGOAT' (aGVHD Onset Anticipation Tianjin) dataset).

aGOAT contained data from 584 adult and 45 pediatric cases, and 16% of the adult cohort and 24% of the pediatric cohort suffered from severe aGVHD (Supplementary Table 1). There was a substantial difference in overall survival between the severe aGVHD cases and the other patients in the adult cohort (Fig. 1a). aGOAT encompassed a total of 194 dynamic variables for the adult cohort and 159 dynamic variables for the pediatric cohort (Supplementary Table 2). The dynamic variables were not measured uniformly across all the patients (Fig. 1b and Supplementary Table 3). Fifteen peri-transplantation variables were also included in aGOAT (Supplementary Table 4).

[1]State Key Laboratory of Experimental Hematology, National Clinical Research Center for Blood Diseases, Institute of Hematology & Blood Diseases Hospital, Chinese Academy of Medical Sciences & Peking Union Medical College, Tianjin, China. [2]Yidu Cloud Technology Inc., Beijing, China. [3]These authors contributed equally: Xueou Liu, Yigeng Cao, Ye Guo, Xiaowen Gong, Yahui Feng, Yao Wang. ✉e-mail: xfzhu@ihcams.ac.cn; jiangerlie@163.com; chenjunren@ihcams.ac.cn

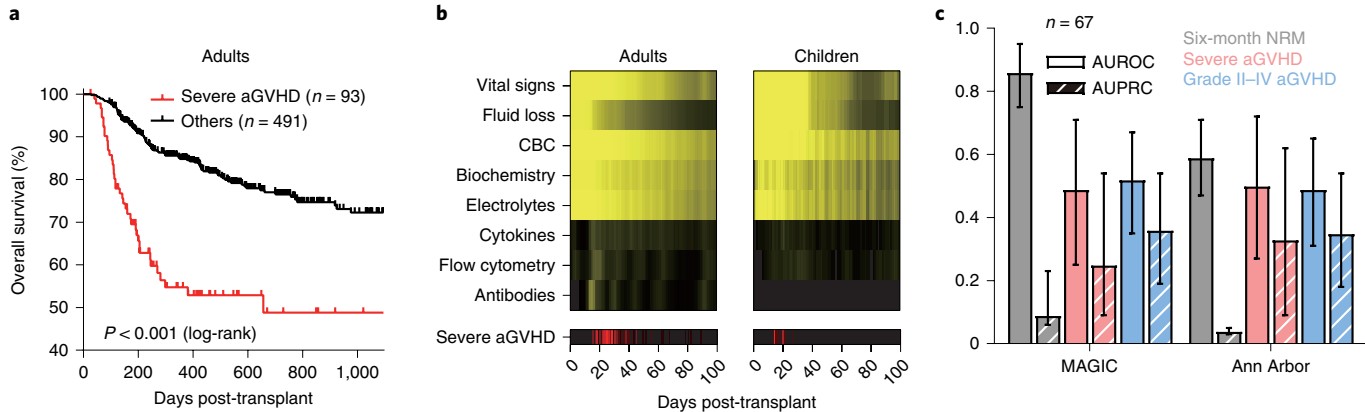

**Fig. 1 | Characteristics of the aGOAT dataset. a**, Overall survival of the adult cohort in the aGOAT dataset. The shown curves are Kaplan–Meier estimators. Red, severe aGVHD cases; black, the other cases. The survival curves for the pediatric cohort are provided in Supplementary Fig. 2. **b**, Data densities and severe aGVHD onset distributions in the aGOAT dataset. Top: data densities of dynamic variables after transplantation in the adult and pediatric cohorts in the aGOAT dataset (after 'time-limited sample-and-hold' data imputations). CBC, complete blood count. Bottom: temporal distributions of severe aGVHD onset. Brighter colors in the heat maps indicate higher densities. The numerical values of missing-data rates for individual dynamic variables are provided in Supplementary Table 3. **c**, Validation of the two-biomarker MAGIC score and three-biomarker Ann Arbor score. Shown here are AUROCs and AUPRCs that were calculated based on the 67 adult patients in the aGOAT dataset who had plasma biomarker values collected at days 6–8. Error bars are 95% CIs based on bootstrapping (1,000 bootstrap samples of the original 67 patients). Solid bars, AUROCs; striped bars, AUPRCs; gray, predicting six-month NRM; pink, predicting severe aGVHD within 100 days; light blue, predicting grade II–IV aGVHD within 100 days.

We have devised a dynamic probabilistic model—'daGOAT' (dynamic aGVHD Onset Anticipation Tianjin)—that integrates multidimensional time-series data to calculate the risk for severe aGVHD. Our model updates the risk score $\varphi_i(t)$ for developing severe aGVHD between $t+1$ and $t+\delta$ according to

$$\varphi_i(t) = \rho(\mathbf{z_i}) + \sum_{\tau \in [t-\delta+1,\, t]} \sum_k (I_{ik\tau}\theta_k\,(x_{ik}(\tau),\, \tau)), \quad (1)$$

where $\rho(\mathbf{z_i})$ and $\theta_k(x_{ik}(\tau), \tau)$ define the contribution of all peri-transplantation features $\mathbf{z_i}$ and the contribution of the individual dynamic variable $x_{ik}(\tau)$, respectively, to the relative risk of the $i$th patient developing severe aGVHD between $t+1$ and $t+\delta$.

The motivation behind equation (1) was to first highlight short-stretch temporal patterns that were suggestive of prodromes for severe aGVHD, with $\delta$ being the length of the moving time window (we set $\delta=14$). The higher the $\varphi_i(t)$, the likelier there was a prodrome between $\tau=t-\delta+1$ and $\tau=t$. Considering the evolving nature of immune reconstitution after HSCT, we assumed the development of severe aGVHD was not a stationary process and defined $\theta_k(\cdot)$ to be a function of time. In other words, the same value for the same clinical feature might have different implications at different times after transplantation. The final risk score on each day was highly dependent on all available clinical information and computed scores in previous days, so, in this sense, daGOAT is an adaptive algorithm.

We compared daGOAT to two landmark-specific plasma bio-marker-based models (the two-biomarker MAGIC score[8] and the three-biomarker Ann Arbor score (based on ST2, Reg3α and tumor necrosis factor receptor 1, TNFR1)[9]), two peri-transplantation features-based models (fitted using Naïve Bayes ('PeriHSCT-NB') or Random Forest ('PeriHSCT-RF')) and XGBoost (a gradient-boosting tree algorithm that permits data missing)[13].

In the adult cohort, plasma biomarker data at days 6–8 were available for 67 patients. The MAGIC score achieved AUROCs of 0.86 and 0.49 for six-month NRM and severe aGVHD, respectively, and the Ann Arbor score achieved AUROCs of 0.59 and 0.50 for six-month NRM and severe aGVHD, respectively (Fig. 1c). The pediatric cohort's biomarker data were too small in size to test the biomarker models.

To evaluate daGOAT, PeriHSCT-NB, PeriHSCT-RF and XGBoost, the patients who received HSCT before (excluding) 1 December 2020 were designated as training sets and the rest of the patients as test sets. Internal validation within the training sets (a series of temporal splitting within the training sets) and holdout validation on the test sets were then conducted.

In the adult cohort, the Q1 (25th percentile), Q2 (median) and Q3 (75th percentile) time points for disease onset were days 24, 29 and 39, respectively. In internal validation ($n=519$), daGOAT's area under the precision-recall curve (AUPRC) peaked at 0.42 (mean; s.d., 0.39 (range, 0.06–1.00)) on day 23, surpassing all the other models (Extended Data Fig. 1a,b). It is worthwhile noting, however, that the PeriHSCT models outcompeted daGOAT during the first two weeks after transplantation. AUROC values showed a trend similar to AUPRC values. daGOAT's AUROC peaked at 0.78 (mean; s.d., 0.17 (range, 0.58–1.00)) on day 23 (Extended Data Fig. 1c and Supplementary Fig. 3). In holdout validation ($n=65$), daGOAT's AUPRC and AUROC peaked at 0.82 and 0.94, respectively, on day 23, outcompeting all the other models (Extended Data Fig. 2a–c and Supplementary Fig. 3). Again, the PeriHSCT models surpassed daGOAT during the first two weeks (Extended Data Fig. 2a and Supplementary Fig. 3). For the adult cohort, on day 23, hazard ratios (HRs) between high-risk (risk score top 1/6) and low-risk (risk score bottom 5/6) patients according to daGOAT were 2.07 (95% confidence interval (CI), 1.03–4.15) and 18.40 (95% CI, 3.95–85.60) in internal and holdout validations, respectively, in the ensuing two-week window (Extended Data Figs. 1d and 2d).

In the pediatric cohort, the Q1, Q2 and Q3 time points for disease onset were days 15, 17 and 21, respectively. In internal validation ($n=39$), daGOAT's AUPRC peaked at 0.75 (mean; s.d. 0.28 (range, 0.45–1.00)) on day 10, surpassing all the other models (Extended Data Fig. 3a,b). AUROC values showed a similar trend to AUPRCs. daGOAT's AUROC peaked at 0.87 (mean; s.d., 0.13 (range, 0.75–1.00)) on day 10 (Extended Data Fig. 3c and Supplementary Fig. 3). In holdout validation ($n=6$), daGOAT's AUPRC and AUROC values were both 1.00 on day 10—on par with XGBoost and outcompeting the PeriHSCT models (Extended Data Fig. 4a–c and Supplementary Fig. 3). Note, however, that XGBoost's AUPRC and AUROC values were more stable in time than daGOAT's in

holdout validation for the pediatric cases (Extended Data Fig. 4a and Supplementary Fig. 3). For the pediatric cohort, on day 10, HRs between patients classified as high-risk (risk score top 1/6) and low-risk (risk score bottom 5/6) identified by daGOAT were 4.47 (95% CI, 1.14–17.50) and incalculable (due to the small sample size) in internal and holdout validations, respectively, in the ensuing two-week window (Extended Data Figs. 3d and 4d).

We also investigated the differential contributions of individual dynamic features to aGVHD prediction. The importance score of a feature at a given time point was calculated as the decremental change of AUPRC (based on internal validation within the training set) at that time point if the feature was entirely ignored from day 1 through day 100. Interestingly, the importance scores of many features varied smoothly with time (that is, scores at neighboring time points were similar to one another; Fig. 2a,b). We conducted an ablation experiment with daGOAT by removing its smoothing component and then testing the truncated version of the model. As expected, daGOAT without smoothing performed worse in both the adult and pediatric cohorts (Fig. 2c).

We ranked all the dynamic features according to their maximum importance scores during days 8–30 (Fig. 2a,b and Supplementary Table 5). Spearman's rank correlation coefficients between feature importance and data density were 0.51 and 0.72 in the adult and pediatric cohorts, respectively. Although data densities were clearly influential in the rankings of the dynamic features, some top-ranked features nonetheless had lower data densities (especially in the adult cohort).

The ability of daGOAT to predict severe aGVHD depended on leveraging the multidimensionality of data (Fig. 2d,e). In the training sets, performance metrics peaked when ~20% top-ranked variables were used in the model. In holdout validations, however, it took at least 80% and 50% of the dynamic features for model performance to approach saturation in the adult and pediatric cohorts, respectively.

To explore plausible explanations for daGOAT's advantage over benchmarks in predicting severe aGVHD, we conducted simulation experiments in which three data characteristic parameters were manipulated systematically. The first was the complexity of the underlying process. A more complex process had a higher number of effector features that each had independent association with event onset. On the other hand, when the underlying process was more simple, most observed features were dummies and made no contribution to relative risk. The second parameter was the smoothness of the underlying process. When the underlying process was 'smooth', each feature's contribution to relative risk could change over time, but there were no rapid up-and-down swings. The third parameter was the data missing rate. Data missing was expected to make model fitting more difficult.

Examining the simulation results, we found that daGOAT outperformed XGBoost when most of the observed variables were associated with event onsets, when the underlying event-generating process was smooth and when there was much data missing (Extended Data Fig. 5a).

Although this study focused on HSCT, we also tested whether our proposed approach could be generalized to a remotely unrelated scenario of dynamic event forecasting using multivariate time-series. More specifically, we asked whether waist-mounted smartphone inertial sensor data could be utilized by daGOAT to anticipate a person's postural change, that is, to predict if a sitting person was about to stand up. A publicly available smartphone inertial sensor dataset[14] was downloaded from the UCI Machine Learning Repository. The dataset contained time-series data ($\Delta t = 1.28$ s) of 30 human subjects that covered 561 features. Each discrete time point was associated with a label that indicated whether the person was sitting, standing up (transitioning from sitting to standing), standing or performing another activity at that moment. There was no missing value. Although we did not have direct insights into the underlying neurobehavioral process, we postulated that the relationship between prodromic subtle motions and human postural change was probably smooth.

Four models—daGOAT, Naïve Bayes, Random Forest and XGBoost—were tested on the smartphone inertial sensor dataset. For daGOAT, a '+' data segment would be akin to a 'severe aGVHD case', and its associated 561-feature time-series would be its 'presymptomatic clinical data'. daGOAT's AUPRC and AUROC peaked at 0.29 and 0.73 (Extended Data Fig. 5b–e), respectively, $\geq 2.56$ s before the observed postural transition, outperforming Naïve Bayes, Random Forest and XGBoost.

Although we caution that our simulation experiments and smartphone data analysis were far from encompassing all possible real-world scenarios, they nonetheless served as a conduit to understanding the possible mechanisms behind daGOAT's comparative advantage in severe aGVHD prediction.

In contrast to modeling in HSCT, machine learning research on dynamic risk monitoring based on high-density multidimensional time-series data has been particularly active in intensive care in recent years. Instead of banking on a small set of biomarkers, researchers have taken a holistic approach that considers time-series of a high number of features to forecast shock[15,16] and to make artificial intelligence-based recommendations for sepsis treatment[17]. Homogeneous ultrahigh data density in intensive care units is nevertheless an outlier situation. As of today, 'spotty data' (inadequate data densities) remain the norm in most real-world healthcare settings (outside of clinical trials), and this includes HSCT. Machine learning research in HSCT is furthermore hampered by smaller sample sizes.

When most of the observed features independently contribute to relative risk, there is little benefit for a model to distinguish between true effectors and dummies. Accordingly, daGOAT does not conduct any variable selection, whereas—despite the small sample size and the time-varying nature of feature contributions to relative risk—on each day XGBoost would have to pick a new set of key variables to grow trees. The comparatively good performance of our modeling approach suggests that it is feasible to predict severe aGVHD cost-effectively when taking a panoramic and

**Fig. 2 | Characteristics of the daGOAT model. a,b,** Temporal profiles of the importance scores for all dynamic features in the adult (**a**) and pediatric (**b**) cohorts. CV, coefficient of variation; DC, dendritic cell; RBC, red blood cell. The importance scores (theoretical range: –1 to 1) were calculated based on internal validation within the training sets. Red, positive importance score; blue, negative importance score. The features were ranked according to their maximum importance scores during days 8–30. The full rankings of all the features are available in Supplementary Table 5. The average data density of each dynamic feature was calculated by dividing its total data volume (that is, the total number of available values after 'time-limited sample-and-hold' data imputations) by the total number of patients and by the total number of days (30 days). Right panels: orange bars, maximum importance score during days 8–30; black lines, average data density during days 1–30. **c,** Ablation experiment with daGOAT. Removing the smoothing component of the algorithm hurt daGOAT's performance at the peak performance days (adults, day 23; children, day 10). There was only one run (comparing 'without smoothing' and 'with smoothing') in each scenario, and the standard error was incalculable. Striped bars, AUPRCs; solid bars, AUROCs; brown, internal validation; green, holdout validation. **d,e,** Relationships between the number of top-ranked features included in the model and the model's performance metrics (AUPRC and AUROC) on the peak performance days (adults, day 23; children, day 10) in internal and holdout validations for the adult (**d**) and pediatric (**e**) cohorts. Crosses, AUPRCs; circles, AUROCs; brown, internal validation; green, holdout validation.

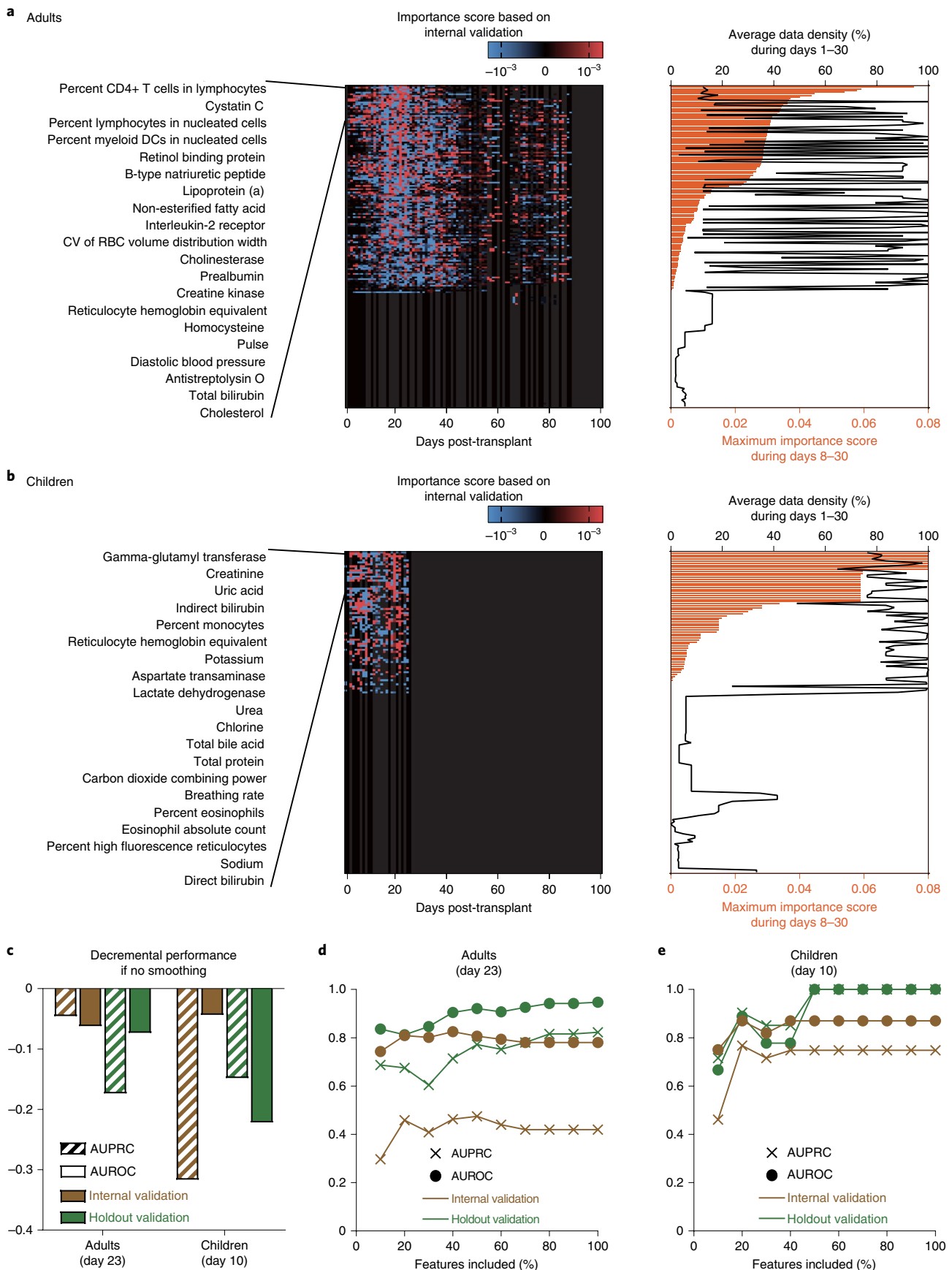

dynamic view of a patient's clinical profile. The average total daily cost (charged to the patient) for data collection from day 1 through day 30 post-transplant to support daGOAT was 261 renminbi per day per pediatric patient and 428 renminbi per day per adult patient at the IHCAMS. Despite the large number of dynamic variables included in our model, most of the data utilized in the daGOAT algorithm are collected in routine clinical care after transplantation and thus do not incur additional cost.

For deployment in clinical settings, the daGOAT model must be integrated into the hospital information system. On any given day we have approximately 100 patients who have recently undergone HSCT and are still hospitalized at the IHCAMS; these are the patients whose dynamic clinical data need to be updated daily. Our semi-automatic data process takes less than 30 min to extract the 100 patients' newly collected data on the latest day from the electronic health records and subsequently append the new incoming data to the data accumulated in previous days. daGOAT is fast to compute. On average, computing $\varphi_i(t)$ for 100 consecutive days for one patient takes ~0.5 s. Model fitting is also reasonably fast. Fitting daGOAT on our adult training set, for example, took less than 1 min using a typical desktop computer. In summary, daGOAT is easy to implement, provided that the hospital information system is sufficiently 'modern'.

Regrettably, this study was limited to data from one hematological center in China, and additional validation at other hospitals will be needed. The ultimate litmus test of our model would be testing whether we can reduce early mortality after transplantation by applying the model prospectively to administer intensified prophylactic immunosuppression to a targeted subset of allo-HSCT patients who are predicted to have high risk for developing severe aGVHD.

## Methods

**The aGOAT dataset.** We focused on modeling severe aGVHD in HLA-mismatched allo-HSCT, because HLA mismatch is the most important factor associated with aGVHD[18]. It should be noted, however, that 97% of the HLA-mismatched adult and 100% of the HLA-mismatched pediatric instances were haploidentical in the final dataset (Supplementary Table 1). In the following, we describe, in detail, the process of compiling the aGOAT dataset (Supplementary Fig. 1).

Post-transplant multidimensional time-series clinical data of 598 adult patients (age >16 years) who received HLA-mismatched allo-HSCT with stem cells sourced from peripheral blood, bone marrow or both between 1 April 2012 and 30 April 2021 and 54 pediatric patients (age ≤16 years) who received HLA-mismatched allo-HSCT with stem cells sourced from peripheral blood, bone marrow or both between 1 April 2018 and 31 March 2021 at the IHCAMS were able to be electronically retrieved and curated.

The medical records for each case were reviewed by two or three physicians to confirm the aGVHD diagnosis and grading (according to the MAGIC criteria[19]). To avoid ambiguity, onset of aGVHD was uniformly defined as the day of initiating aGVHD treatment. After the physicians' review, 16 cases (ten adults and six children) were eliminated due to failure of neutrophil engraftment within 30 days of transplantation. An additional seven cases (four adults and three children) were eliminated because the recorded date of neutrophil engraftment (defined as the date of the first of three consecutive measurements spanning ≥3 days of achieving a sustained peripheral blood neutrophil count of >$500 \times 10^6 l^{-1}$) did not precede the recorded onset of aGVHD.

The final dataset contained 584 adult cases and 45 pediatric cases. The adult and pediatric cohorts had substantially different baseline distributions in age, primary diseases, stem cell sources, conditioning regimens and aGVHD prophylaxis regimens (Supplementary Table 1). Because the adult cohort and the pediatric cohort were treated at different divisions of the IHCAMS, our modeling efforts on the two cohorts were two independent validations of the daGOAT algorithm.

Sixteen percent of the adult cohort and 24% of the pediatric cohort suffered from severe aGVHD within 100 days. Moreover, the severe aGVHD instances in the pediatric cohort tended to experience onset much earlier than those in the adult cohort (Fig. 1b and Supplementary Table 1). There was a substantial difference in three-year all-cause mortality between the patients with severe aGVHD and the other patients in the adult cohort (HR 3.30 (95% CI, 2.28–4.79); $P < 0.001$, log-rank test) (Fig. 1a), and a similar trend also appeared to exist in the pediatric cohort, although it did not reach statistical significance (HR 4.40 (95% CI, 0.58–33.47); $P = 0.120$, log-rank test) (Supplementary Fig. 2).

aGOAT encompassed a total of 194 dynamic variables for the adult cohort and 159 dynamic variables for the pediatric cohort, collected during the first 100 days after transplantation (Supplementary Table 2), including vital signs, daily fluid loss (due to diarrhea, vomiting and so on), complete blood counts, blood chemistry and electrolytes, peripheral blood/bone marrow immune cell profiles (measured by flow cytometry), plasma inflammatory factor levels and so on. The dynamic variables were not measured uniformly across all patients. Some dynamic variables such as vital signs were available nearly daily, whereas others such as blood immune cell profiles and plasma inflammatory factor levels were measured less frequently and not in all patients. In addition, 15 peri-transplantation variables were also included in aGOAT (Supplementary Table 4), including information related to primary disease, blood type, stem cell source, HLA mismatch, conditioning regimen before transplantation, use of antithymocyte globulin in conditioning, aGVHD prophylaxis regimen, transplantation year and so on.

Outlier values in vital signs (for example, exorbitant values for body temperature) were made blank. Whenever a dynamic variable was measured more than once (distinct samples) on one particular day for one patient, the average measurement value of that day was used for that day for that patient. We applied the 'time-limited sample-and-hold' approach commonly used in intensive care unit data analysis[17] to augment the aGOAT dataset (holding time set to three days after sampling), based on the hypothesis that most measurements were valid for three additional days. This augmented dataset was still very sparse in multiple categories of dynamic variables (Fig. 1b and Supplementary Table 3). No other missing-data imputation procedure was conducted to address the problem of nonuniform data measurement.

**Validation of the MAGIC score and the Ann Arbor score.** The MAGIC score was calculated as $-11.263 + 1.844(\log(ST2)) + 0.577(\log(REG3\alpha))$. The Ann Arbor score was calculated as $-9.169 + 0.598(\log(TNFR1)) - 0.028(\log(REG3\alpha)) + 0.189(\log(ST2))$. All the used coefficient values were identical to those reported in the original reports[8,9]. The original reports did not specify the units for plasma biomarker measurements, and the two scores used different bases for the logarithm (MAGIC, 10; Ann Arbor, 2). None of these, however, affected the AUROC and AUPRC calculations.

**daGOAT model.** We designed the daGOAT algorithm with the motivation to leverage one presumed nature of post-HSCT time-series data: the underlying biological process for aGVHD onset is multidimensional and smooth with respect to time. By explicitly taking the temporal order of data into account, daGOAT borrows strengths from neighboring time points. Even if a feature is missing a value on one particular day, the model might still learn its contribution to relative risk on that day by interpolating between neighboring time points.

Our model integrates multidimensional time-series data to calculate risk for severe aGVHD onset between $t + 1$ and $t + \delta$ according to

$$\varphi_i(t) = \rho(\mathbf{z}_i) + \sum_{\tau \in [t-\delta+1, t]} \sum_k (I_{ik\tau} \theta_k(x_{ik}(\tau), \tau)),$$

where $\rho(\mathbf{z}_i)$ and $\theta_k(x_{ik}(\tau), \tau)$ define the contribution of all peri-transplantation features $\mathbf{z}_i$ and the contribution of the individual dynamic variable $x_{ik}(\tau)$, respectively, to the relative risk of patient $i$ developing severe aGVHD between $t + 1$ and $t + \delta$. $I_{ik\tau} = 0$ when $x_{ik}(\tau)$ lacks value for the $i$th patient, and $I_{ik\tau} = 1$ otherwise. Although equation (1) does not presume the size of the time step, updating $\varphi_i(t)$ daily was deemed sufficient for severe aGVHD prediction.

We fitted daGOAT as follows. First, $\rho(\mathbf{z}_i)$ was set to be the log-odds ratio computed by the standard Naïve Bayes algorithm. Second, for every $k$ and $t$, we computed the cutoff value $c_{kt}$ that maximized Shannon's mutual information between the $k$th dynamic variable at time $t$ and severe aGVHD occurrence, then we set $l_k$ and $u_k$ to be the 25th and 75th percentile values among $c_{k1}, \ldots, c_{kT}$, respectively. This step computed the optimal cutoff values $\{l_k, u_k\}$ to discretize the $k$th dynamic variable. Third, for every $k$ and $t$ we computed

$$\rho_{1kt}^{(L)} = P(x_k(t) < l_k | \text{Severe aGVHD onset between } t + 1 \text{ and } t + \delta),$$

$$\rho_{1kt}^{(H)} = P(x_k(t) > u_k | \text{Severe aGVHD onset between } t + 1 \text{ and } t + \delta),$$

$$\rho_{0kt}^{(L)} = P(x_k(t) < l_k | \text{No severe aGVHD onset between } t + 1 \text{ and } t + \delta), \text{ and}$$

$$\rho_{0kt}^{(H)} = P(x_k(t) > u_k | \text{No severe aGVHD onset between } t + 1 \text{ and } t + \delta),$$

then we computed $\hat{\rho}_{1k}^{(L)}(t)$, $\hat{\rho}_{1k}^{(H)}(t)$, $\hat{\rho}_{0k}^{(L)}(t)$ and $\hat{\rho}_{0k}^{(H)}(t)$ as 'smoothed' versions of $\rho_{1kt}^{(L)}$, $\rho_{1kt}^{(H)}$, $\rho_{0kt}^{(L)}$ and $\rho_{0kt}^{(H)}$, respectively, by smoothing-spline fitting (smooth with respect to $t$). This step computed the discretized probability distribution of the $k$th dynamic variable that was smooth along the time axis. Note that we did not conduct interpolations on the raw data. True distributions of feature values in normal,

prodromic and diseased states are unknown, and it is unclear how to best perform interpolations on the raw data for a wide range of variables expressed in various units (for example, should we interpolate values in the original scale or in the log scale?). Instead, our model was distribution-agnostic and performed interpolation on the estimated relative risk contribution terms, that is, in the probability space.

Finally, we defined

$$
\theta_k(x, t) = \begin{cases} \log\left(\frac{\max\left\{0, \hat{\rho}_{1k}^{(L)}(t) + \gamma\right\}}{\max\left\{0, \hat{\rho}_{0k}^{(L)}(t) + \gamma\right\}}\right) & \text{if } x < l_k \\[2ex] \log\left(\frac{\max\left\{0, 1 - \hat{\rho}_{1k}^{(L)}(t) - \hat{\rho}_{1k}^{(H)}(t) + \gamma\right\}}{\max\left\{0, 1 - \hat{\rho}_{0k}^{(L)}(t) - \hat{\rho}_{0k}^{(H)}(t) + \gamma\right\}}\right) & \text{if } l_k \leq x \leq u_k \\[2ex] \log\left(\frac{\max\left\{0, \hat{\rho}_{1k}^{(H)}(t) + \gamma\right\}}{\max\left\{0, \hat{\rho}_{0k}^{(H)}(t) + \gamma\right\}}\right) & \text{if } u_k < x \end{cases}
$$

where $\gamma \geq 0$ is a hyperparameter that we set to be 0.1. When the model was run in the 'no smoothing' mode, $\hat{\rho}_{1k}^{(L)}(t)$, $\hat{\rho}_{1k}^{(H)}(t)$, $\hat{\rho}_{0k}^{(L)}(t)$ and $\hat{\rho}_{0k}^{(H)}(t)$ were not calculated, and $\rho_{1kt}^{(L)}$, $\rho_{1kt}^{(H)}$, $\rho_{0kt}^{(L)}$ and $\rho_{0kt}^{(H)}$ were used instead for calculating $\theta_k(x,t)$.

**Data generation for simulation experiments.** We generated multidimensional time-series data using a simplified version of equation (1):

$$
\varphi_i = \sum_{t \in [1, T]} \sum_{k \in [1, p]} \theta_k(x_{ik}(t), t) \tag{2}
$$

where $\varphi_i$ is the relative risk of the event of interest happening to individual $i$, $p = 50$ and $T = 14$. To further simplify the simulations (without hurting generalizability), we assumed each $x_{ik}(t)$ term had only two possible values: low and high.

First, for each individual $i$ and feature $k$, a smooth time-series $x_{ik}(t)$ was generated by a random walk that meandered between low and high (transition rates: 'low→low' or 'high→high', 0.7; 'low→high' or 'high→low', 0.3).

Second, we simulated values for $\theta_k(x,t)$, the underlying process for event generation. When the underlying process was designated to be more complex (that is, a high percentage of observed variables were indeed correlated with event onset), we set $\theta_k(x,t) = 0$ for $k \in [46, 50]$. On the other hand, when the underlying process was designated to be simple (that is, most observed variables were dummies), we set $\theta_k(x,t) = 0$ for $k \in [6, 50]$. For all the other $k$ values, when the underlying process was designated to be smooth, we first generated a smooth time-series $\alpha_k(t)$ through a random walk with multiplicative Gaussian steps (mean = 1; s.d. = 0.05) starting from $\alpha_k(1) = 1$. On the other hand, when the underlying process was designated to be not smooth, $\alpha_k(t)$ at each $t$ was independently drawn from a uniform distribution between 0 and 1. The values of $\alpha_k(t)$ were then linearly rescaled so that $\min_{t \in [1, T]} \alpha_k(t) = 0.3$ and $\max_{t \in [1, T]} \alpha_k(t) = 0.7$; then we defined $\theta_k(\text{high}, t) = \log\left(\frac{\alpha_k(t)}{1-\alpha_k(t)}\right)$ and $\theta_k(\text{low}, t) = -\theta_k(\text{high}, t)$.

Third, the generated values $x_{ik}(t)$ and $\theta_k(x,t)$ were then plugged into equation (2) to calculate $\varphi_i$. For each $i$, draw a random number $\gamma$ from a uniform distribution between 0 and 1. If $\gamma < \varphi_i$, the event of interest happened to individual $i$; otherwise, the event did not happen to individual $i$.

Finally, when the data missing rate was designated to be high, 60% of the $x_{ik}(t)$ terms were randomly marked as missing; otherwise, there was no data missing.

For each combination of data characteristic parameters, the simulation was repeated 150 times. Each run was conducted with $n = 1,000$ (80:20 random split for holdout validation) and a constant 15% percentage of positive cases within the simulated sample (a larger cohort of individuals was first generated and then randomly downsampled to the desired size).

**The smartphone-based recognition of human activities and postural transitions dataset.** The waist-mounted smartphone inertial sensor dataset[14] was downloaded from the UCI Machine Learning Repository (https://archive.ics.uci.edu/ml/datasets/Smartphone-Based+Recognition+of+Human+Activities+and+Postural+Transitions). Before download, the smartphone dataset had already been randomly partitioned into two subsets, with 70% of the subjects designated as the training set and the rest as the test set. From this dataset we extracted all 8.96-s time-series segments in which a person was continuously sitting and then classified each segment according to whether the person stood up within the next 5.12 s ('+') or not ('−'). The training set contained 789 data segments ('+', $n = 92$; '−', $n = 697$), and the test set contained 292 segments ('+', $n = 40$; '−', $n = 252$).

**Reporting Summary.** Further information on research design is available in the Nature Research Reporting Summary linked to this Article.

## Data availability

Although there was no genetic polymorphism, gene expression or protein sequence data involved in this study, sharing of substantial clinical data generated from China's human genetic resources needs to abide by the Regulations of the People's Republic of China (PRC) on the Administration of Human Genetic Resources. A 'minimum dataset' for severe aGVHD that would be necessary to verify the research in this Article would include all dynamic features (from day 1 through day 100 post-HSCT) listed in Supplementary Table 2, all peri-transplantation features listed in Supplementary Table 4, presence/absence of severe aGVHD within 100 days, and onset dates of severe aGVHD. On 21 February 2022, the PRC Human Genetic Resources Administration Office approved the compilation of a desensitized version of the 'minimum aGOAT dataset' for facilitating international research collaborations (Reference No. CJ0272 (2022)). At the time of the publication of the manuscript, the authors' application to deposit this desensitized dataset at the PRC National Genomics Data Center (NGDC) database was still under review. A mock-up dataset that can be used for demo runs of the daGOAT algorithm is available in a public Zenodo repository (https://doi.org/10.5281/zenodo.6050675)[20]. After publication of this work, the Chinese government has approved the archiving of the aGOAT dataset at the PRC National Genomics Data Center (NGDC) in April, 2022 (ref. no. 2022BAT1224). Accordingly, the authors have made the dataset publicly accessible at the NGDC website (https://ngdc.cncb.ac.cn/omix/release/OMIX001095/). The waist-mounted smartphone inertial sensor dataset is available from the UCI Machine Learning Repository (https://archive.ics.uci.edu/ml/datasets/Smartphone-Based+Recognition+of+Human+Activities+and+Postural+Transitions). Source data for Figs. 1 and 2 and Extended Data Figs. 1–5 are provided with this paper.

## Code availability

The R code used in this study is available in a public GitHub repository at https://github.com/chenjunren-ihcams/daGOAT (https://doi.org/10.5281/zenodo.6041841)[21].

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

## Acknowledgements

This work was supported in part by the State Key Laboratory of Experimental Hematology research grant no. Z20-01 (to J.R.C.) and the CAMS Innovation Fund for Medical Sciences grant no. 2020-I2M-C&T-B-089 (to Y.G.). We thank H. Zhang for assistance in determining neutrophil engraftment dates. The Article processing charge of this manuscript is paid by the Tianjin Science and Technology Plan grant no. 20ZYJDSY00010.

## Author contributions

J.C., E.J. and X. Zhu supervised the study. J.C., E.J. and X. Zhu designed the study, with contributions from Y.G. and Y.C. X.L. coordinated the study, with contributions from S.Z. and Z.S. Y.C., Y.G., M.W., L.Z., X.C., S.F., M.H., E.J. and X. Zhu contributed to data collection. X.L., Y.C., Y.G., X.G., Y.F., M.W. and W.G. compiled, reviewed and curated the dataset, with contributions from N.Z., X.S., X. Zheng and X.C. J.C. designed the algorithm, with contributions from Y.W., M.C. and L.L. J.C., Y.W. and M.C. performed the computation, with contributions from Y.F., X.G., Q.S. and X.L. J.C. wrote the first draft of the manuscript, with contributions from Y.W., X.L., M.C., X.G., Y.C., E.J. and X. Zhu. J.C., Y.W., M.C., X.G., Y.F. and X.L. revised the manuscript, with contributions from M.W., Y.C., Y.G., E.J. and X. Zhu.

## Competing interests

The authors declare no competing interests.

## Ethics declarations

This retrospective study was initiated in October 2020 and became part of a larger-scope research program (NICHE-GOAT), which was approved by the IHCAMS Clinical Research Academic Committee on 11 January 2021 (IIT2021006) and by the IHCAMS Ethics Committee on 7 February 2021 (IIT2021006-EC-1). To avoid biased healthcare or research decisions, patients who received HSCT later than 1 December 2020 were not included in this study until after 7 February 2021. All the patients included in this study signed an informed consent form that permitted their biological samples or data to be utilized for research.

## Additional information

**Extended data** is available for this paper at https://doi.org/10.1038/s43588-022-00213-4.

**Correspondence and requests for materials** should be addressed to Xiaofan Zhu, Erlie Jiang or Junren Chen.

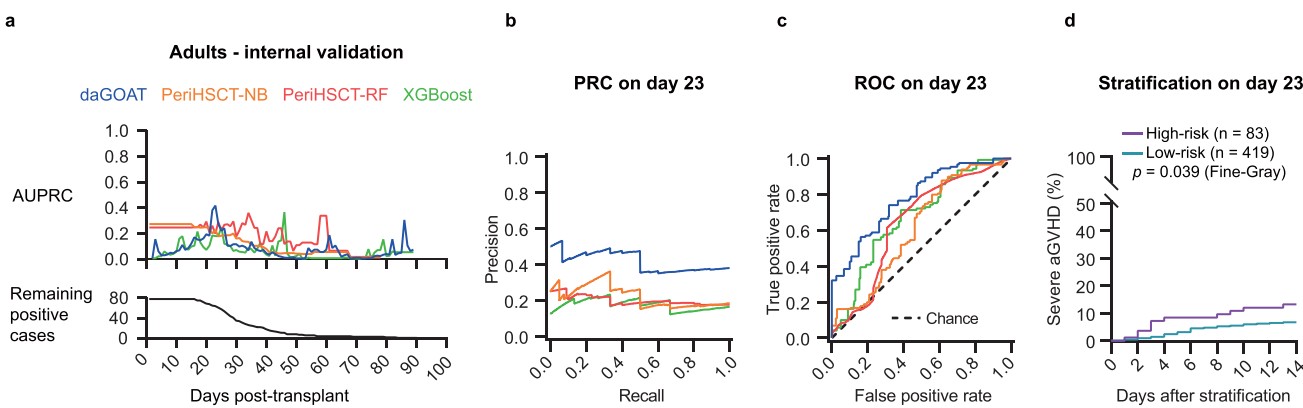

**Extended Data Fig. 1 | Internal validation of daGOAT, PeriHSCT-NB, PeriHSCT-RF, and XGBoost in the adult cohort. a**, Temporal profiles of AUPRC from day 1 through day 100. (Blue, daGOAT; orange, PeriHSCT-NB; red, PeriHSCT-RF; green, XGBoost.) The bottom panel shows how the number of remaining positive cases (severe aGVHD cases that had not had onset) decreased over time in the training set. **b**, Precision-recall curves for the four models on day 23 (daGOAT's peak performance day) in the training set. **c**, Receiver operating characteristic curves for the four models on day 23 in the training set. (The dotted line represents the null model.) **d**, Cumulative incidence curves for severe aGVHD within the ensuing two-week window (days 24–37) after the training-set patients were stratified on day 23. All-cause death was treated as a competing event that precluded severe aGVHD when calculating the cumulative incidence curves for severe aGVHD. 'High-risk' (purple), daGOAT score among top 1/6; 'low-risk' (cyan), daGOAT score among bottom 5/6.

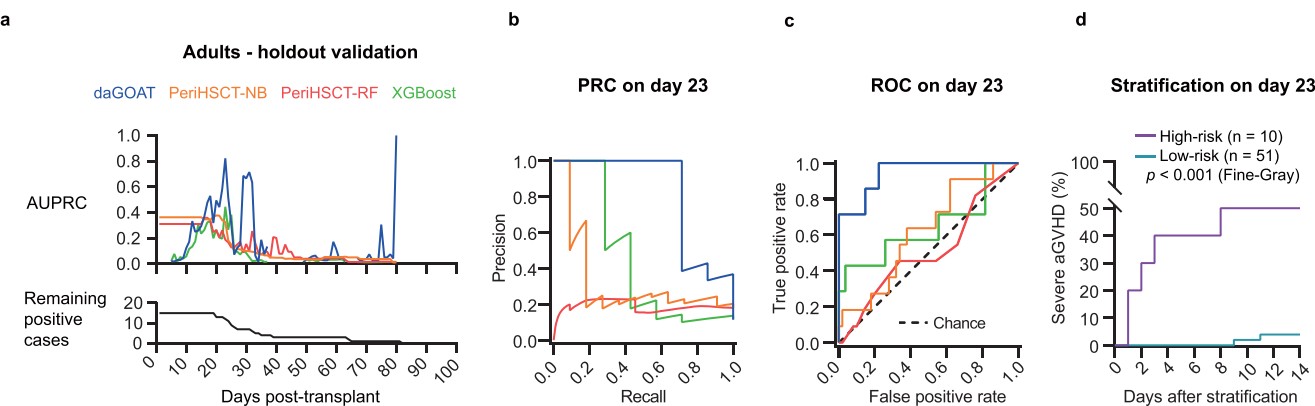

**Extended Data Fig. 2 | Holdout validation of daGOAT, PeriHSCT-NB, PeriHSCT-RF, and XGBoost in the adult cohort. a**, Temporal profiles of AUPRC from day 1 through day 100. The bottom panel shows how the number of remaining positive cases (severe aGVHD cases that had not had onset) decreased over time in the test set. (Blue, daGOAT; orange, PeriHSCT-NB; red, PeriHSCT-RF; green, XGBoost.) **b**, Precision-recall curves for the four models on day 23 (daGOAT's peak performance day for the adults according to internal validation) in the test set. **c**, Receiver operating characteristic curves for the four models on day 23 in the test set. (The dotted line represents the null model.) **d**, Cumulative incidence curves for severe aGVHD within the ensuing two-week window (days 24–37) after the test-set patients were stratified on day 23. All-cause death was treated as a competing event that precluded severe aGVHD when calculating the cumulative incidence curves for severe aGVHD. 'High-risk' (purple), daGOAT score among top 1/6; 'low-risk' (cyan), daGOAT score among bottom 5/6.

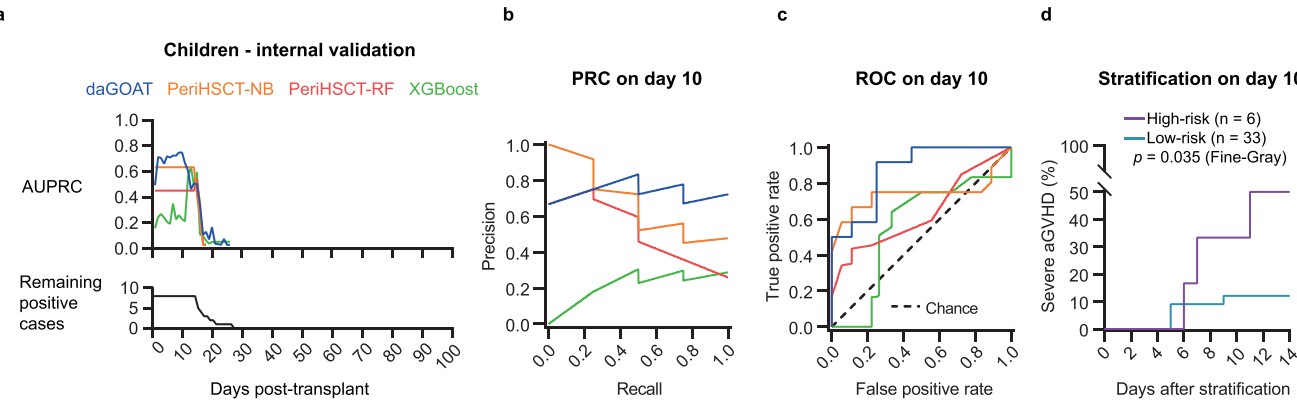

**Extended Data Fig. 3 | Internal validation of daGOAT, PeriHSCT-NB, PeriHSCT-RF, and XGBoost in the pediatric cohort. a**, Temporal profiles of AUPRC from day 1 through day 100. The bottom panel shows how the number of remaining positive cases (severe aGVHD cases that had not had onset) decreased over time in the training set. (Blue, daGOAT; orange, PeriHSCT-NB; red, PeriHSCT-RF; green, XGBoost.) **b**, Precision-recall curves for the four models on day 10 (daGOAT's peak performance day) in the training set. **c**, Receiver operating characteristic curves for the four models on day 10 in the training set. (The dotted line represents the null model.) **d**, Cumulative incidence curves for severe aGVHD within the ensuing two-week window (days 11–24) after the training-set patients were stratified on day 10. All-cause death was treated as a competing event that precluded severe aGVHD when calculating the cumulative incidence curves for severe aGVHD. 'High-risk' (purple), daGOAT score among top 1/6; 'low-risk' (cyan), daGOAT score among bottom 5/6.

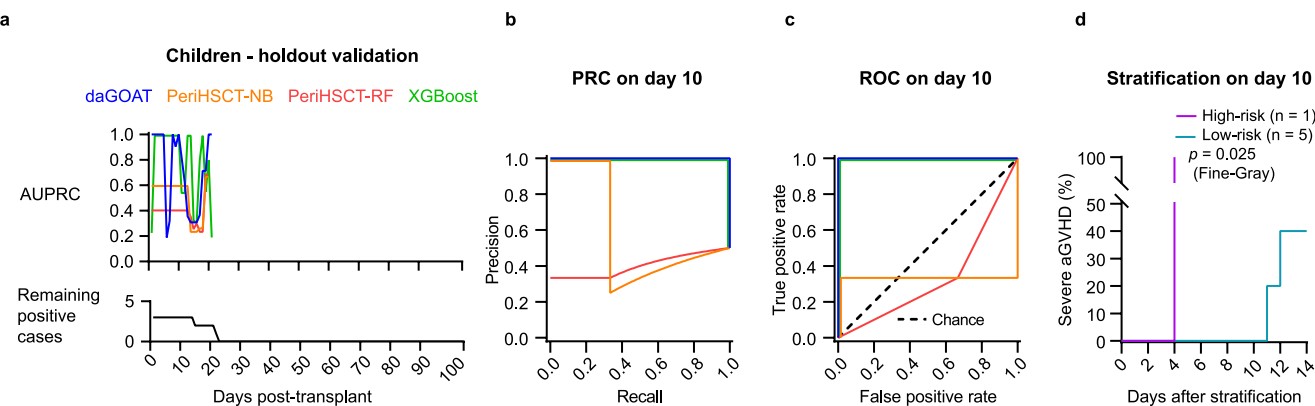

**Extended Data Fig. 4 | Holdout validation of daGOAT, PeriHSCT-NB, PeriHSCT-RF, and XGBoost in the pediatric cohort. a**, Temporal profiles of AUPRC from day 1 through day 100. The bottom panel shows how the number of remaining positive cases (severe aGVHD cases that had not had onset) decreased over time in the test set. (Blue, daGOAT; orange, PeriHSCT-NB; red, PeriHSCT-RF; green, XGBoost.) **b**, Precision-recall curves for the four models on day 10 (daGOAT's peak performance day for the pediatric cases according to internal validation) in the test set. **c**, Receiver operating characteristic curves for the four models on day 10 in the test set. (The dotted line represents the null model.) **d**, Cumulative incidence curves for severe aGVHD within the ensuing two-week window (days 11–24) after the test-set patients were stratified on day 10. All-cause death was treated as a competing event that precluded severe aGVHD when calculating the cumulative incidence curves for severe aGVHD. 'High-risk' (purple), daGOAT score among top 1/6; 'low-risk' (cyan), daGOAT score among bottom 5/6.

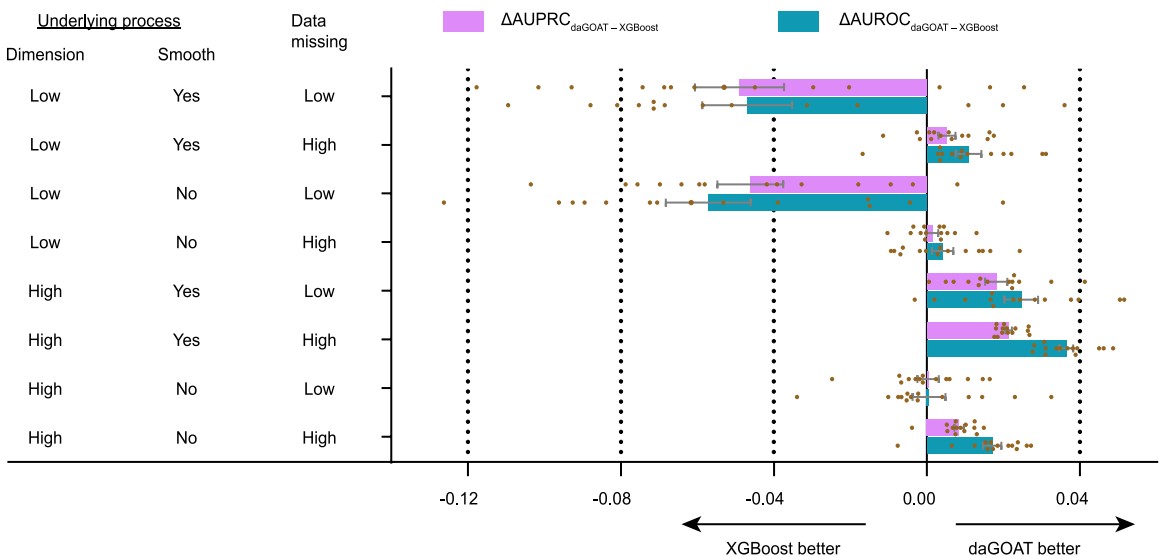

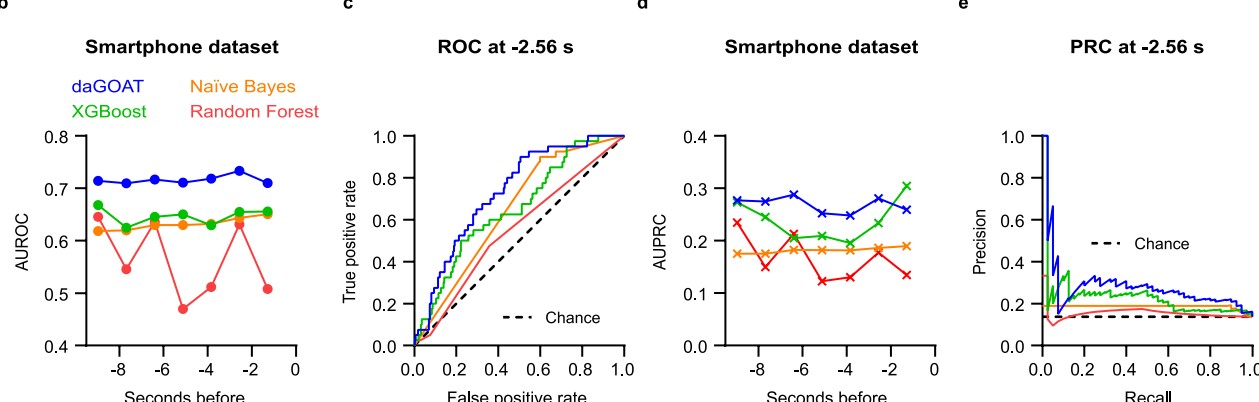

**Extended Data Fig. 5 | Extension experiments on the daGOAT algorithm. a**, Performance of daGOAT and XGBoost in short time-series data simulation experiments under various scenarios of data characteristics. The length of the simulated time-series was uniformly T =14. Mean values and standard errors of ΔAUROCs and ΔAUPRCs across the 14 time points are shown here, along with the raw values at the 14 time points overlaid as dots. (Purple, ΔAUPRC; cyan, ΔAUROC.) **b – e**, Performance of daGOAT, XGBoost, Naïve Bayes, and Random Forest on the UCI Machine Learning Repository smartphone inertial sensor data. Temporal profiles for AUROC (**b**) and AUPRC (**d**) show that daGOAT outperformed the other models from −8.0 to −2.5 s before postural transition. Receiver operating characteristic curves and precision-recall curves at −2.56 s (**c,e**) are also shown here to compare the models in better detail; dotted lines represent null models. (Blue, daGOAT; green, XGBoost; orange, Naïve Bayes; red, Random Forest.)

# Reporting Summary

Nature Research wishes to improve the reproducibility of the work that we publish. This form provides structure for consistency and transparency in reporting. For further information on Nature Research policies, see our Editorial Policies and the Editorial Policy Checklist.

## Statistics

For all statistical analyses, confirm that the following items are present in the figure legend, table legend, main text, or Methods section.

| n/a | Confirmed | |
|---|---|---|
| ☐ | ☒ | The exact sample size (*n*) for each experimental group/condition, given as a discrete number and unit of measurement |
| ☐ | ☒ | A statement on whether measurements were taken from distinct samples or whether the same sample was measured repeatedly |
| ☐ | ☒ | The statistical test(s) used AND whether they are one- or two-sided *Only common tests should be described solely by name; describe more complex techniques in the Methods section.* |
| ☐ | ☒ | A description of all covariates tested |
| ☐ | ☒ | A description of any assumptions or corrections, such as tests of normality and adjustment for multiple comparisons |
| ☐ | ☒ | A full description of the statistical parameters including central tendency (e.g. means) or other basic estimates (e.g. regression coefficient) AND variation (e.g. standard deviation) or associated estimates of uncertainty (e.g. confidence intervals) |
| ☐ | ☒ | For null hypothesis testing, the test statistic (e.g. *F*, *t*, *r*) with confidence intervals, effect sizes, degrees of freedom and *P* value noted *Give P values as exact values whenever suitable.* |
| ☒ | ☐ | For Bayesian analysis, information on the choice of priors and Markov chain Monte Carlo settings |
| ☒ | ☐ | For hierarchical and complex designs, identification of the appropriate level for tests and full reporting of outcomes |
| ☐ | ☒ | Estimates of effect sizes (e.g. Cohen's *d*, Pearson's *r*), indicating how they were calculated |

*Our web collection on statistics for biologists contains articles on many of the points above.*

## Software and code

Policy information about availability of computer code

| Data collection | All data used in this study were retrieved from the electronic health record system at the Institute of Hematology, Chinese Academy of Medical Sciences (IHCAMS). Custom scripts used for accessing the hospital's internal database and for curating the retrieved data have to remain confidential. |
|---|---|
| Data analysis | R code used in this study is available in a public GitHub repository at https://github.com/chenjunren-ihcams/daGOAT (DOI: 10.5281/zenodo.6041841). Utilized R libraries and versions: readxl (ver. 1.3.1), dplyr (ver. 1.0.8), e1071 (ver. 1.7-9), randomForest (ver. 4.7-1), xgboost (ver. 1.5.0.2), pROC (ver. 1.18.0), and PRROC (ver. 1.3.1). |

For manuscripts utilizing custom algorithms or software that are central to the research but not yet described in published literature, software must be made available to editors and reviewers. We strongly encourage code deposition in a community repository (e.g. GitHub). See the Nature Research guidelines for submitting code & software for further information.

## Data

Policy information about availability of data

All manuscripts must include a data availability statement. This statement should provide the following information, where applicable:

- Accession codes, unique identifiers, or web links for publicly available datasets
- A list of figures that have associated raw data
- A description of any restrictions on data availability

While there was no genetic polymorphism, gene expression, or protein sequence data involved in this study, sharing of substantial clinical data generated from China's human genetic resources needs to abide by the Regulations of the People's Republic of China (PRC) on the Administration of Human Genetic Resources. A 'minimum dataset' for severe aGVHD that would be necessary to verify the research in this article would include all dynamic features (from day 1 through day 100 post-HSCT) listed in Supplementary Table 2, all peri-transplantation features listed in Supplementary Table 4, presence/absence of severe aGVHD within 100 days,

April 2020

and onset dates of severe aGVHD. On 21 February 2022, the PRC Human Genetic Resources Administration Office approved the compilation of a desensitized version of the 'minimum aGOAT dataset' for facilitating international research collaborations (Reference No. CJ0272 (2022)). At the time of the publication of the manuscript, the authors' application to deposit this desensitized dataset at the PRC National Genomics Data Center (NGDC) database is still under review. To inquire the latest status of public accessibility to the desensitized 'minimum aGOAT dataset' at the NGDC database, please contact Junren Chen (chenjunren@ihcams.ac.cn). A mock-up dataset that can be used for demo runs of the daGOAT algorithm is available in a public Zenodo repository at https://zenodo.org/record/6050675#.Ygcg1N_P2Uk (DOI: 10.5281/zenodo.6050675). The waist-mounted smartphone inertial sensor dataset is available from the UCI Machine Learning Repository (https://archive.ics.uci.edu/ml/datasets/Smartphone-Based+Recognition+of+Human+Activities+and+Postural+Transitions). Source data for Figs. 1 and 2 and Extended Data Figs. 1–5 are provided with this paper.

# Field-specific reporting

Please select the one below that is the best fit for your research. If you are not sure, read the appropriate sections before making your selection.

☒ Life sciences ☐ Behavioural & social sciences ☐ Ecological, evolutionary & environmental sciences

For a reference copy of the document with all sections, see nature.com/documents/nr-reporting-summary-flat.pdf

# Life sciences study design

All studies must disclose on these points even when the disclosure is negative.

| | |
|---|---|
| Sample size | Post-transplant multidimensional time-series clinical data of 584 adult patients (age >16) who received HLA-mismatched allo-HSCT with stem cells sourced from peripheral blood, bone marrow, or both between 1 April 2012 and 30 April 2021 and 45 pediatric patients (age ≤16) who received HLA- mismatched allo-HSCT with stem cells sourced from peripheral blood, bone marrow, or both between 1 April 2018 and 31 March 2021 at the IHCAMS were able to be electronically retrieved and curated. |
| Data exclusions | 16 cases (10 adults and 6 children) were eliminated due to failure of neutrophil engraftment within 30 days of transplantation. Additional 7 cases (4 adults and 3 children) were eliminated, because the recorded date of neutrophil engraftment (defined as 'the date of the first of three consecutive measurements spanning ≥3 days of achieving a sustained peripheral blood neutrophil count of >500×10^6/L') did not precede the recorded onset of aGVHD. |
| Replication | The same modeling approach was independently applied to both the adult and pediatric cohorts, and in both cohorts both internal validation and holdout validation were independently performed. Our main finding – i.e., daGOAT outperformed the benchmarks – was replicated in all four scenarios (adult-internal validation, adult-holdout validation, children-internal validation, children-holdout validation). |
| Randomization | Not applicable. There was no experiment performed on any human subject. This study was a retrospective analysis on patients treated at the IHCAMS. |
| Blinding | Not applicable. There was no experiment performed on any human subject. This study was a retrospective analysis on patients treated at the IHCAMS. |

# Reporting for specific materials, systems and methods

We require information from authors about some types of materials, experimental systems and methods used in many studies. Here, indicate whether each material, system or method listed is relevant to your study. If you are not sure if a list item applies to your research, read the appropriate section before selecting a response.

## Materials & experimental systems

| n/a | Involved in the study |
|---|---|
| ☒ | Antibodies |
| ☒ | Eukaryotic cell lines |
| ☒ | Palaeontology and archaeology |
| ☒ | Animals and other organisms |
| ☐ | ☒ Human research participants |
| ☒ | Clinical data |
| ☒ | Dual use research of concern |

## Methods

| n/a | Involved in the study |
|---|---|
| ☒ | ChIP-seq |
| ☒ | Flow cytometry |
| ☒ | MRI-based neuroimaging |

# Human research participants

Policy information about studies involving human research participants

| | |
|---|---|
| Population characteristics | The adult cohort contained 584 patients with the following characteristics: age 17-64 (median 35), male sex 58%, primarily myeloid neoplasms (AML (44%), ALL (22%), MDS (20%), aplastic anemia (9%)), and receiving primarily (97%) haploidentical hematopoietic stem cell transplantation.

The pediatric cohort contained 45 patients with the following characteristics: age 2-15 (median 9), male sex 69%, primarily |

bone marrow failure diseases (47%) and myeloid neoplasms (49%), and all (100%) receiving haploidentical hematopoietic stem cell transplantation.

Recruitment          629 consecutive cases whose data could be retrieved were included in this study. 23 cases were excluded from this study during QA/QC (see 'Data exclusions' on p. 2).

Ethics oversight     This retrospective study was approved by the IHCAMS Clinical Research Academic Committee on 11 January 2021 (IIT2021006) and by the IHCAMS Ethics Committee on 7 February 2021 (IIT2021006-EC-1). All the patients included in this study signed an informed consent form that permitted their biological samples or data to be utilized for research.

Note that full information on the approval of the study protocol must also be provided in the manuscript.

