## [Peer Review File · Nature Computational Science]

Peer Review Information

Journal: Nature Computational Science

Manuscript Title: Dynamic forecasting of severe acute graft-versus-host disease after transplantation

Corresponding author name(s): Xiaofan Zhu, Erlic Jiang, Junren Chen

Reviewer Comments & Decisions:

Decision Letter, initial version:

Date: 7th January 22 14:43:59

Last Sent: 7th January 22 14:43:59

Triggered By: Kaitlin McCardle

From: kaitlin.mccardle@us.nature.com

To: chenjunren@ihcams.ac.cn

BCC: kaitlin.mccardle@us.nature.com

Subject: Decision on Nature Computational Science manuscript NATCOMPUTSCI-21-0881

Message: ** Please ensure you delete the link to your author homepage in this e-mail if you wish to forward it to your co-authors. **

Dear Mr CHEN,

Your manuscript "Dynamic forecasting of severe acute graft-versus-host disease after transplantation" has now been seen by 3 referees, whose comments are appended below. You will see that while they find your work of interest, they have raised points that need to be addressed before we can make a decision on publication.

The referees' reports seem to be quite clear. Naturally, we will need you to address all of the points raised.

While we ask you to address all of the points raised, the following points need to be substantially worked on:

- Provide additional comparisons with state-of-the-art approaches, as well as validate the results produced by daGOAT
- Demonstrate the generalizability of your approach

- Demonstrate (perhaps with experiments) the broad implications of this work
- Better explain the methodology and justify the reasons for choosing these techniques
- Better explain the random forest results in the main text
- Expand on the MAGIC biomarkers study and compare your results to previous publications that show the usability of MAGIC biomarkers when combined.
- Further explain how this can be properly used in a clinical setting.

Please note that should your paper eventually be accepted, all code related to experiments and methodology will need to be deposited into a proper repository.

Please use the following link to submit your revised manuscript and a point-by-point response to the referees' comments (which should be in a separate document to any cover letter):

[REDACTED]

** This url links to your confidential homepage and associated information about manuscripts you may have submitted or be reviewing for us. If you wish to forward this e-mail to co-authors, please delete this link to your homepage first. **

To aid in the review process, we would appreciate it if you could also provide a copy of your manuscript files that indicates your revisions by making use of Track Changes or similar mark-up tools. Please also ensure that all correspondence is marked with your Nature Computational Science reference number in the subject line.

In addition, please make sure to upload a Word Document or LaTeX version of your text, to assist us in the editorial stage.

To improve transparency in authorship, we request that all authors identified as 'corresponding author' on published papers create and link their Open Researcher and Contributor Identifier (ORCID) with their account on the Manuscript Tracking System (MTS), prior to acceptance. ORCID helps the scientific community achieve unambiguous attribution of all scholarly contributions. You can create and link your ORCID from the home page of the MTS by clicking on 'Modify my Springer Nature account'. For more information please visit www.springernature.com/orcid.

We hope to receive your revised paper within three weeks. If you cannot send it within this time, please let us know.

Best regards,

Kaitlin McCardle
Editor
Nature Computational Science

Reviewers comments:

Reviewer #1 (Remarks to the Author):

In the present manuscript by Liu et al., the authors address the prediction of severe graft-versus-host disease (GVHD) after allogeneic hematopoietic stem cell transplantation with dynamic forecasting using a nonparametric approach that accommodates heterogeneous data. The model was trained and tested on an adult- and pediatric transplant cohort receiving predominantly haploidentical transplantation and resulted in a predictive value of up to 0.7 as measured by the AUROC. The authors also compare the predictive capacity of a baseline model using static parameters, a random forest ML model and established GVHD biomarker models and reveal limitations in the use of these biomarkers. Finally, the authors apply their forecasting model to a different use case and dataset structure in the anticipated prediction of movements in subjects monitored by wearable devices. Indeed, there is a need to address predictions in medicine timely with dynamic models, which may integrate many variables over time. The authors present results from a reasonably sized adult patient cohort with 599 patients receiving alternative donor transplantation from haploidentical donors in the modern transplant era after 2010. The cohort is quite young as compared to other real world studies including patients with hematologic malignancies from the US or Europe. The incidence of severe aGVHD was comparable to existing studies for the adults and quite high for the children. The CONSORT diagram provides an overview of the patient selection. Data availability and Ethics sections are transparent. From a superficial review the R code was clean, corresponding to the formula in the methods section and is potentially executable (not tested). The study is of potential interest; however, it has major limitations, which need to be addressed by the authors to improve the manuscript.

Major:

- Given the current manuscript structure, the question remains what core message the authors would like to convey. Is it a proof-of-principle study of non-parametric forecasting using two kinds of heterogeneous datasets or do they want to address the complex issue of forecasting GVHD? In its present form each issue did not receive sufficient detail. This manuscript should focus on one of these: If the authors chose to reveal the feasibility of their approach in GVHD the additional data could potentially fit into the supplement. Then, the GVHD specific data and its discussion and contextualization should be extended. If the authors chose to point to the use of this prediction technique in two diverse datasets with very limited human cases (n) and a high number of features this could be an alternative strategy. The cost-effectiveness section should also be reconsidered under this question.
- Dynamic forecasting is indeed a matter of interest in medical care and has the potential to improve outcome assessment. However, the validity of AUROC has a limitations in this setting and should be completed by the use of areas under the precision-recall curves (AUPRCs) for both events and samples, which respond better in the context of repeated real time assessments at different timepoints (see also: Hyland SL, Nature Medicine 2020). For clarity, the AUROC curves should also be displayed as figures.
- The presented AUROC of 0.7 could be an improvement to published existing static prediction models, yet the inclusion of only 11 static baseline parameters might have

weakened the baseline model as comparator, it should be tested if the data used for this baseline model could be enlarged. There was no separate internal validation of this model, which should be added via cohort splitting in training and validation cohort rather than only by 5x cross-validation. In the pediatric cohort, the Random Forest resulted in comparable AUROC to the daGOAT model, which was shown in the reporting summary but not sufficiently discussed in the manuscript.

- The comparison to the MAGIC biomarkers needs to be evolved further. In this Haplo cohort the MAGIC biomarkers do not have sufficient predictive value for severe aGVHD, at least when analyzed separately. This is a finding of potential major relevance and interest, but it needs to be solid. There are several publications (e.g. Major-Monfried Blood 2018 or Srinagesh Blood 2019) that have demonstrated the validity of the MAGIC biomarkers, in particular when combined. The authors should discuss these findings e.g. in the context of their Haplo HCT strategy used. Is there a particularity about this specific cohort that could explain this difference? Would a combination improve the results? The authors should also show their MAGIC biomarker results in the prediction of all-grade GVHD.

- Overall GVHD and severe GVHD data should also be shown as cumulative incidence curves with competing events (Fine and Gray 1999). The clinical data on cumulative incidence of relapse and non-relapse mortality should also be added including competing events. The inclusion of GVHD events at least up to day +100 would enable the comparison to other transplant studies in the field.

Minor:

- The scales of the figure axes (e.g. Figure 1a, 1f, 1g are not homogenous. I would prefer to change this as it may irritate the reader in particular to compare the results between the adult and pediatric cohort. The timescale of 1e+f should end, at least, at day+ 100 as these patients may still develop GVHD, not at day +40. Same for the scales of the axes of Fig 1 c and d. Same for the Fig. 2a+b+e.
- The different axes scales make comparisons and conclusions difficult to follow and to review e.g. line 163 f. "Second, daGOAT's learning curve rose considerably faster in the pediatric cohort than in the adult cohort (Fig. 2d)."
- The introduction does not discuss potential weaknesses of the MAGIC biomarkers with sufficient detail as needed for an open research question
- This Haploidentical cohort should not be primarily denominated as HLA mismatched, which points to e.g. 9/10 or 8/10 alleles HLA matching. In this context, the statement in the methods line 210 "We focused on modeling severe aGVHD in HLA-mismatched allo-HSCT, because HLA mismatch is the most important factor associated with aGVHD" is a bit misleading.
- The conditioning and immunosuppressive strategy in the adult and pediatric cohorts is not detailed and should be displayed in the supplement.
- The supplemental table of abbreviated features used in the model is not intuitively clear and must also be listed as standard units.
- The descriptive statistic on subgroup of the early severe GVHD cases does not really differ from all GVHD cases in time of onset, this subgroup's role is not entirely clear
- Label 1a e.g. "others" instead of "otherwise"
- The main text does not discriminate between introduction and results sections. The introduction should not refer to own results. Figure 3 is not presented in the results

section.

- The broadness of the included features without any selection is a strength but also a weakness of the model. Feature selection and tuning could potentially improve the predictive capacity, if an ML approach was adopted.
- The grouped features as heat map did not differentiate well between feature importance. To better understand the model's performance, it would be of interest to show the importance of individual features to highlight which features contributed most to this prediction and if this contribution changed over time.
- There are several language and style issues through the manuscript e.g. line 104 "was the contribution of 'dynamic' variable" better: "defined the contribution of 'dynamic' variables"

Reviewer #2 (Remarks to the Author):

The paper presents an innovative approach (daGOAT) for dynamic forecast of severe acute graft-versus-host-disease (aGVHD) risk among adult and pediatric patients by considering dynamic (i.e., time varying) as well as stationary variables. The risk scores are updated dynamically for each patient based on the data changes. This study is an excellent example of utilizing dynamic clinical variables for determining the risk for the onset of aGVHD in transplant patients. However, there are basic issues that need to be addressed and clarified. Below are my comments and suggestions for improvements:

1. Authors claimed that their proposed approach is non-parametric with real-time risk score updation, however it's not clear how the proposed approach is real-time? What is the time frequency for considering the approach as real-time, for example- is it considered to be minutely, hourly, or daily updation of scores?
Since the approach is real-time, it must be adaptive too, which is not mentioned anywhere in the paper, it's good to clarify on the adaptiveness of the approach i.e. how the approach is adaptive and it's effect on the outcomes.
2. Authors mentioned - Attempts that tried to use biomarkers measured prior to the appearance of aGVHD signs to forecast severe aGVHD gave conflicting results (Lines 54-57). This claim is not clear after looking through the referred papers (7,10,11,13,14). Please provide more clarification on this, how the results are conflicting?
3. It will be great if authors provide more descriptive statistics on the missingness of data. How much data is missing along with it's distribution? A table to show it will be better.
4. Authors proposed dynamic probabilistic model daGOAT that integrated multidimensional time series data to compute risk. Risk scores are updated daily. Need more intuitive explanation of modelling technique (Eqn. 1), it is not clear how the authors come up with the model, is it the extension of an existing approach or based on an existing approach.
5. Authors proposed method is validated with the performance of stationary features only model, landmark specific plasma biomarker levels and landmark-specific random survival forest models [Lines 115- 118]. There is no comparison of the proposed approach with the state-of-the-art non-parametric approaches.

Why does the proposed method is not compared to other state-of-the-art non-parametric approaches? More experiments on the state-of-the-art approaches using aGOAT dataset and their comparison with the proposed approach will be better.

Reviewer #3 (Remarks to the Author):

This is a good model for predicting time-dependent outcome like aGVHD after allo-HSCT.

This reviewer wonder how this technics can really be applied to the clinics, i.e. this predictive score has instantly impacts on the patient therapeutic procedures. At least the mechanism to calculate the score more easily is mandatory. Please discuss more on this point.

This is the machine learning dependent predictive model combining patient background characteristics and ongoing laboratory data with clinical courses (data of multiple time points). This type of model, compared to the one using only patient background characteristics (data of one time), can be a more accurate and clinically relevant model.

In this context, this Reviewer acknowledges the impacts of this model, and understands it as a good model in this field.

As mentioned, this can be a good model as a methodology and their paper is successful as a proof-of-concept study. However, the authors focused on the very minute part of the biology (extremely rare); post-transplant complication for hematological malignancies. The Reviewer is not sure their methodology can be applied to more generalized topics (ie. the analysis of more common diseases, like hypertension, strokes, or some cancers), which make the reviewer suggest that this paper should be submitted to the other more specific journal (that covers hematology or transplantation) not to the general journal of Nature computation science.

This paper may aim to apply their methods to the actual clinical field (bedsides of the patient), and to be helpful in the clinical decision. However, their calculation methods using complicated computing systems seem to be too complicated. That is why the method to calculate the scores which they invented more easily looks necessary.

Author Rebuttal to Initial comments

Response to referees

#	Editor's comment	Response
0.01	Provide additional comparisons with state-of-the-art approaches, as well as validate the results produced by daGOAT	We choose to benchmark against XGBoost (Chen & Guestrin, 2016), a state-of-the-art tree boosting algorithm that permits missing data (Figs. 1 & 3). Moreover, we now include the results of both internal validation (within training sets) and holdout validation (using test sets collected after 1 December 2020) in the revised manuscript (lines 159–195 and Fig. 1).
0.02	Demonstrate the generalizability of your approach	The main reason we were not persuasive in demonstrating the generalizability of daGOAT in our previous manuscript – we realized – is that we ourselves did not understand when and why daGOAT worked. We have since conducted a series of simulation experiments (lines 224–246, lines 505–540, and Figs. 3a) to interrogate daGOAT under different data situations and reached the conclusion that daGOAT is more suited for tackling a ‘large p , small n ’ time-series problem when the underlying process for event generation is smooth and multidimensional and when many features are missing values at multiple time points. Fig. 1b, Figs. 2a,b, and Supplementary Table 3 clearly showed that the aGOAT dataset falls under this category.
0.03	Demonstrate (perhaps with experiments) the broad implications of this work	
0.04	Better explain the methodology and justify the reasons for choosing these techniques	Intuitions behind the daGOAT algorithm are now first briefed in Results (lines 132-144) and then further expanded in Discussion (lines 338-357).
0.05	Better explain the random forest results in the main text	Inspired by Reviewer #1's comments, we have modified the daGOAT algorithm to make it fully dynamic for forecasting events. The new daGOAT forecasts on each day if severe aGVHD will happen in the ensuing two-week window. AUROC peaked at ≥ 0.78 in both the adult and the pediatric cohorts right before most severe aGVHD cases were about to experience onsets.

		Furthermore, following Reviewer #2's suggestion – we now use a state-of-the-art benchmark algorithm, XGBoost, in the revised manuscript. Random Survival Forests (RSF) – one of the benchmark algorithms we used previously – is no longer considered, for three reasons: 1) Unlike XGBoost, RSF requires data imputation; 2) RSF requires lengthy model-fitting time; and 3) we are no longer doing survival analysis.
0.06	Expand on the MAGIC biomarkers study and compare your results to previous publications that show the usability of MAGIC biomarkers when combined	We tested both the two-biomarker MAGIC score (Hartwell et al. (JCI Insight 2017)) and the three-biomarker Ann Arbor score (Levine et al. (Lancet Haematol 2015)) on our dataset. Both scores failed to predict severe aGVHD in our study cohorts (lines 153-156, lines 459-465, and Fig. 1c).
0.07	Further explain how this can be properly used in a clinical setting	Embedding daGOAT in the hospital information system would allow the risk scores to be computed daily for each hospitalized patient. See lines 391-406 for a more detailed description in Discussion.

#	Reviewer's comment	Response
1.01	Given the current manuscript structure, the question remains what core message the authors would like to convey. Is it a proof-of-principle study of non-parametric forecasting using two kinds of heterogenous datasets or do they want to address the complex issue of forecasting GVHD? In its present form each issue did not receive sufficient detail. This manuscript should focus on one of these: If the authors chose to reveal the feasibility of their approach in GVHD the additional data could potentially fit into the supplement. Then, the GVHD specific data and its discussion	We thank Reviewer #1 for enlightening us and also for encouraging us to think hard about the positioning of our paper. It is of utmost importance that we tell the story of aGVHD convincingly. As such, we have adopted all technical and stylistic suggestions made by Reviewer #1 so that we can be fully transparent when communicating to our worldwide colleagues in blood diseases. We still plan to publicize our dataset to facilitate continuing research in this area (pending governmental approval), and we still believe our discussion of healthcare costs is relevant—at least for physicians and patients in developing countries. We, however, also acknowledge the need to appeal to non-hematologists when publishing in Nature Computational Science. It was only after reading Reviewer #1's comment that we realized both the

	and contextualization should be extended. If the authors chose to point to the use of this prediction technique in two diverse datasets with very limited human cases (n) and a high number of features this could be an alternative strategy. The cost-effectiveness section should also be reconsidered under this question.	aGVHD dataset and the smartphone dataset we analyzed were ‘large p , small n ’. This prompted us to systematically conduct a series of simulation experiments to investigate why and when daGOAT works. We hope this ‘excursion’ would help non-hematologists appreciate the kind of problems our algorithm is suited for.
1.02	Dynamic forecasting is indeed a matter of interest in medical care and has the potential to improve outcome assessment. However, the validity of AUROC has a limitations in this setting and should be completed by the use of areas under the precision-recall curves (AUPRCs) for both events and samples, which respond better in the context of repeated real time assessments at different timepoints (see also: Hyland SL, Nature Medicine 2020). For clarity, the AUROC curves should also be displayed as figures.	In this study, we focused on predicting first-time onset of severe aGVHD for each patient. (Recurrence of severe aGVHD is of course possible, but it is often hard to rule out that the second episode is simply a continuation of the first episode, especially when the second episode happens after titration of immunosuppression treatment.) Trend curves of AUROCs and AUPRCs for predictions at the event level from day 1 through day 100 are now reported in Fig. 1 and Supplementary Fig. 3.
1.03	The presented AUROC of 0.7 could be an improvement to published existing static prediction models, yet the inclusion of only 11 static baseline parameters might have weakened the baseline model as comparator, it should be tested if the data used for this baseline model could be enlarged. There was no separate internal validation of this model, which should be added via cohort splitting in training and	We have expanded the set of peri-HSCT features to include also: GVHD prophylaxis, use of antithymocyte globulin in conditioning, HLA mismatch, and transplantation year (lines 115–119 and Supplementary Table 4). We now use 1 December 2020 as the cut-off date to divide both the adult and pediatric cohorts into training sets and test sets. Both internal validation within the training sets and holdout validation on the test sets were then conducted. See lines 159-195, Fig. 1, and Figs. 2c–e. In the previous manuscript, AUROC (0.71) of

	validation cohort rather than only by 5x cross-validation. In the pediatric cohort, the Random Forest resulted in comparable AUROC to the daGOAT model, which was shown in the reporting summary but not sufficiently discussed in the manuscript.	Random Survival Forests (RSF) was on par with the previous version of daGOAT in the pediatric cohort. We have since modified the daGOAT algorithm, and the new daGOAT performs considerably better (AUROC ≥ 0.87) in the pediatric cohort. We no longer use RSF as a benchmark; rather, we now benchmark against XGBoost. The reporting summary has been updated accordingly.
1.04	The comparison to the MAGIC biomarkers needs to be evolved further. In this Haplo cohort the MAGIC biomarkers do not have sufficient predictive value for severe aGVHD, at least when analyzed separately. This is a finding of potential major relevance and interest, but it needs to be solid. There are several publications (e.g. Major-Monfried Blood 2018 or Srinagesh Blood 2019) that have demonstrated the validity of the MAGIC biomarkers, in particular when combined. The authors should discuss these findings e.g. in the context of their Haplo HCT strategy used. Is there a particularity about this specific cohort that could explain this difference? Would a combination improve the results? The authors should also show their MAGIC biomarker results in the prediction of all-grade GVHD.	Most biomarker papers actually did not study the prediction of aGVHD (using biomarker data collected before aGVHD); rather, they often studied the prediction of treatment response or long-term clinical outcome after aGVHD onset (using biomarker data collected at or after aGVHD onset)—some prominent examples are: Vander Lugt et al. (N Engl J Med 2013), Levine et al. (Lancet Haematol 2015), Major-Monfried et al. (Blood 2018), and Srinagesh et al. (Blood Adv 2019). McDonald et al. (Blood 2015) studied both; in their Table 5, they reported the AUROC of using ST2 measured at days 11 to 17 to predict severe aGVHD (peak grade 3-4 (n = 11) vs grade 0-2), and it was 0.56. The two-biomarker MAGIC model by Hartwell et al. (JCI Insight 2017) was designed for predicting 6-month NRM using ST2 and Reg3alpha measured on day 7. Predicting NRM, however, is not the same as predicting severe aGVHD or even aGVHD-caused NRM. Hartwell et al. did not report the AUROC for predicting severe aGVHD or aGVHD-caused NRM. Nonetheless, using the data they provided in Table S6, we found that the proportion of NRM cases that were attributed to aGVHD (using the number of NRM cases as the denominator) was statistically indistinguishable between MAGIC score-stratified ‘high-risk’ and ‘low-risk’ groups (64% (41/64) vs. 57% (47/83); p-value 0.458, chi-squared test). One cannot rule out that the higher aGVHD mortality (using the total number of cases as the denominator)

		in their high-risk group was a mere reflection of higher overall NRM in the high-risk group. While both Major-Monfried et al. (2018) and Srinagesh et al. (2019) reported to have validated the MAGIC model, neither of these two studies used the model to predict aGVHD; rather, they creatively repurposed the MAGIC score for predicting treatment response after aGVHD onset (using biomarker data collected at or after aGVHD onset). Interestingly, if one examines Hartwell et al. (2017) closely, one finds that – even for forecasting NRM – the MAGIC score performs better when measured at GVHD onset (their Fig. 3A) than on day 7 (their Figs. 1A-C). Review of previous biomarker studies is now expanded in Introduction (lines 56-72). In preparation of this revision, we have tested both the two-biomarker MAGIC score (Hartwell et al. (2017)) and the three-biomarker Ann Arbor score (Levine et al. (2015)) on our dataset (aGOAT). We found that, consistent with our understanding of the published literature, the two-biomarker MAGIC score could predict 6-month NRM (AUROC 0.86), but both the MAGIC score and the Ann Arbor score failed to predict either severe aGVHD or grade II-IV aGVHD (AUROCs ≈ 0.5). See lines 153-156, lines 459-465, and Fig. 1c. Similar to McDonald et al. (2015), we did not analyze grade I aGVHD. We cannot speak for McDonald et al., but our reason is that clinically it is hard to diagnose grade I aGVHD with full certainty.
1.05	Overall GVHD and severe GVHD data should also be shown as cumulative incidence curves with competing events (Fine and Gray 1999). The clinical data on cumulative	Requested information is now included in Supplementary Table 1 and Supplementary Fig. 2 . Also, cumulative incidence curves in Figs. 1g,k,o,s are now plotted with all-cause death as the competing event.

	incidence of relapse and non-relapse mortality should also be added including competing events. The inclusion of GVHD events at least up to day +100 would enable the comparison to other transplant studies in the field.	All statistical analyses of aGVHD, including its modeling, now extend up to day 100 (see – for instance – Fig. 1).
1.06	The scales of the figure axes (e.g. Figure 1a, 1f, 1g are not homogenous. I would prefer to change this as it may irritate the reader in particular to compare the results between the adult and pediatric cohort. The timescale of 1e+f should end, at least, at day+ 100 as these patients may still develop GVHD, not at day +40. Same for the scales of the axes of Fig 1 c and d. Same for the Fig. 2a+b+e.	Most figures are now plotted up to day 100 (Figs. 1 & 2). Figs. 1g,k,o,s are exceptions, because we now forecast severe aGVHD within a two-week moving window.
1.07	The different axes scales make comparisons and conclusions difficult to follow and to review e.g. line 163 f. “Second, daGOAT’s learning curve rose considerably faster in the pediatric cohort than in the adult cohort (Fig. 2d).”	Comparisons between the adult and the pediatric cohorts are now plotted in the same scale. We now focus on analyzing the impact of variable selection on model performance (lines 217–222 and Figs. 2d,e), and as such learning curves have been removed from the manuscript. That more patients lead to better model-fitting is neither interesting nor surprising.
1.08	The introduction does not discuss potential weaknesses of the MAGIC biomarkers with sufficient detail as needed for an open research question	Review of previous biomarker studies is now expanded in Introduction (lines 56-72).
1.09	This Haploidentical cohort should not be primarily denominated as HLA mismatched, which points to e.g. 9/10 or 8/10 alleles HLA matching. In this context, the	We thank Reviewer #1 for calling this out. We did not set out to study haploidentical HSCT; it just happened that nearly all of our non-cord blood HLA-mismatched cases were haploidentical (there were cord blood cases in the pediatric cohort), as we

	statement in the methods line 210 “We focused on modeling severe aGVHD in HLA-mismatched allo-HSCT, because HLA mismatch is the most important factor associated with aGVHD” is a bit misleading.	reported previously in Supplementary Table 1. We have remedied our data cleaning/description issues as follows: First, we removed all the pediatric cord blood cases (n=37) from the dataset. Second, lines 422–426 now read: “We focused on modeling severe aGVHD in HLA-mismatched allo-HSCT, because HLA mismatch is the most important factor associated with aGVHD. It should be pointed out that 97% of the HLA-mismatched adult cases and 100% of the HLA-mismatched pediatric cases were haploidentical in the final dataset (Supplementary Table 1).” A similar clarification was given at the very beginning of the Results section (lines 98-99). Third, in the process of data cleaning for this revision, we noticed we had previously made the mistake of assuming all haploidentical cases were HLA-mismatched. It turned out 16 adult haplo-HSCT cases were actually HLA-matched, and they are now excluded from the dataset. Fourth, previously we only modeled aGVHD up to day 40 and excluded one adult case from the dataset because the patient died before day 40. Now, since we analyze severe aGVHD in the context of competing events, there is no longer need to exclude this case. Our final dataset contains 584 adult cases and 45 pediatric cases. The CONSORT diagram has been updated accordingly (Supplementary Fig. 1).
1.10	The conditioning and immunosuppressive strategy in the adult and pediatric cohorts is not detailed and should be	Requested information is now included in Supplementary Table 1 and utilized for fitting the peri-HSCT features-based models (Fig. 1).

	displayed in the supplement.	
1.11	The supplemental table of abbreviated features used in the model is not intuitively clear and must also be listed as standard units.	Strange-looking variable names are from the legacy data at the IHCAMS. We keep the original variable names, but each variable name is now accompanied by an explanation in Supplementary Table 2 . Rather than determining what ‘standard units’ are and worrying about numerical errors caused by unit conversions, we include the units we use at the IHCAMS—also in Supplementary Table 2 .
1.12	The descriptive statistic on subgroup of the early severe GVHD cases does not really differ from all GVHD cases in time of onset, this subgroup's role is not entirely clear	Figs. 1 & 2 and Supplementary Table 1 are fixed now.
1.13	Label 1a e.g. "others" instead of "otherwise"	Fixed.
1.14	The main text does not discriminate between introduction and results sections. The introduction should not refer to own results. Figure 3 is not presented in the results section.	Introduction has undergone a major rework. Fig. 3 is now presented in the Results section.
1.15	The broadness of the included features without any selection is a strength but also a weakness of the model. Feature selection and tuning could potentially improve the predictive capacity, if an ML approach was adopted.	We calculated the importance score of a feature at a given time point as the decremental change of AUPRC (based on internal validation within the training set) at that time point if the feature was entirely ignored from day 1 through day 100. All the dynamic features were then ranked according to their maximum important scores during days 8 to 30 (Figs. 2a,b and Supplementary Table 5). Using only top-ranked features in the model was evidently advantageous in internal validations of the training sets, requiring only $\approx 20\%$ of the variables. In holdout validations, however, substantially more variables were needed to ensure better predictive performance; it took at least 80% and 50% of the dynamic features for model performance to approach saturation in the adult and the pediatric cohorts,

		respectively (Fig. 2d,e).
1.16	The grouped features as heat map did not differentiate well between feature importance. To better understand the model's performance, it would be of interest to show the importance of individual features to highlight which features contributed most to this prediction and if this contribution changed over time.	Figs. 2a,b have undergone a major rework. Fig. 1b is still plotted in grouped features, because features in the same group have very similar data densities.
1.17	There are several language and style issues through the manuscript e.g. line 104 "was the contribution of 'dynamic' variable" better: "defined the contribution of 'dynamic' variables"	We need help. We have fixed all language and style issues that had been pointed out to us.
2.01	Authors claimed that their proposed approach is non-parametric with real-time risk score updation, however it's not clear how the proposed approach is real-time? What is the time frequency for considering the approach as real-time, for example- is it considered to be minutely, hourly, or daily updation of scores?	Eqs. (1) and (2) do not presume the size of time step. In the context of severe aGVHD prediction, we deem it sufficient to update risk scores daily. This decision is primarily driven by the temporal density of collected data in the post-HSCT period. The following sentence is inserted at lines 141–144 : "Although Eq. (1) did not presume the size of time step, updating $\varphi_i(t)$ daily was deemed sufficient for severe aGVHD prediction (later in a separate application of the daGOAT model, we would use a much shorter time step)." We acknowledge that the phrase 'real-time' in our previous manuscript was misleading. We have removed this phrase from the entire manuscript.
2.02	Since the approach is real-time, it must be adaptive too, which is not mentioned anywhere in the paper, it's good to clarify on the adaptiveness of the approach i.e. how the approach is adaptive and it's effect on the outcomes.	Our model is adaptive in two senses: First, the computed score on each day is highly dependent on all available clinical information and computed scores in previous days. Second, we do not assume the underlying biological

		process is stationary, and therefore $\theta_k(\cdot)$ is defined to be a function of time. In other words, the same value for the same clinical feature might have different implications at different times after transplantation. Lines 132–141 have been inserted in the Results section to better explain how the concept of adaptiveness is incorporated into the daGOAT algorithm.
2.03	Authors mentioned - Attempts that tried to use biomarkers measured prior to the appearance of aGVHD signs to forecast severe aGVHD gave conflicting results (Lines 54-57). This claim is not clear after looking through the referred papers (7,10,11,13,14). Please provide more clarification on this, how the results are conflicting?	For instance, it was reported in at least some studied cohorts that ST2 had either no significant association with aGVHD (Solan et al. (Front Immunol 2019), tables 2 & 3) or a low AUROC (0.56) for forecasting severe aGVHD (McDonald et al. (Blood 2015), table 5) if measured at days 11 to 17 post-transplant (i.e., ‘before’ aGVHD onset). On the other hand, one study of a Japanese cohort did show association between grade II–IV aGVHD and ST2 measured on day 14 (AUROC 0.66) (Matsumura et al. (Turk J Haematol 2020), Fig. S1). Surprisingly few biomarker papers actually did study the prediction of aGVHD (using biomarker data collected before aGVHD); rather, they often studied the prediction of treatment response or long-term clinical outcome after aGVHD onset (using biomarker data collected at or after aGVHD onset)—two notable examples published in high-profile journals are: Vander Lugt et al. (N Engl J Med 2013) and Levine et al. (Lancet Haematol 2015). Hartwell et al. (JCI Insight 2017) – the original MAGIC score paper – primarily focused on predicting 6-month non-relapse mortality (NRM) using biomarkers measured on day 7 (i.e., before aGVHD onset) (see their Figs. 1 & 2)—note, however, that their Fig. 3 was about application of the MAGIC algorithm at aGVHD onset. Predicting NRM, nevertheless, is not the same as predicting

		aGVHD or aGVHD mortality. Although in their Fig. S5 Hartwell et al. showed higher GVHD mortality in their high-risk group, we believe this was due to higher overall NRM in the high-risk group. One can use the data Hartwell et al. provided in Table S6 to calculate that aGVHD was only slightly more likely to be a cause of death for the NRM cases in high-risk group (41/64=64%) than for the NRM cases in the low-risk group (47/83=57%), and this difference did not pass chi-squared test (p-value = 0.458). When we tested the MAGIC score on our aGOAT dataset, we confirmed that the MAGIC score was indeed predictive of 6-month NRM (AUROC 0.86), but it was not predictive of severe aGVHD (AUROC 0.49). We now provide a more detailed review of biomarker literature in Introduction (lines 56–72) and also report our validation result of the MAGIC score in Fig. 1c.
2.04	It will be great if authors provide more descriptive statistics on the missingness of data. How much data is missing along with it's distribution? A table to show it will be better.	Requested information is now available in Supplementary Table 3.
2.05	Authors proposed dynamic probabilistic model daGOAT that integrated multidimensional time series data to compute risk. Risk scores are updated daily. Need more intuitive explanation of modelling technique (Eqn. 1), it is not clear how the authors come up with the model, is it the extension of an existing approach or based on an existing approach.	We thank Reviewer #2 for reminding us to verbalize our thinking process better. Motivations behind the daGOAT algorithm are now detailed in lines 132–141 and lines 338–357: We had originally designed the daGOAT algorithm with the motivation to leverage one presumed nature of post-HSCT time-series data: the underlying biological process for aGVHD onset is multidimensional and smooth with respect to time. By explicitly taking temporal order of data into account, daGOAT borrows strengths between neighboring time

		points. Even if a feature is missing value on one particular day, the model might still learn its contribution to relative risk on that day by interpolating between neighboring time points. daGOAT, however, does not conduct interpolations on the raw data. True distributions of feature values in normal, prodromic, and diseased states are unknown, and it is unclear how to best perform interpolations on the raw data for a wide range of variables expressed in various units (for instance, should we interpolate values in the original scale or in the log scale?). Instead, daGOAT is distribution-agnostic and performs interpolation on the estimated relative risk contribution terms, i.e., in the probability space. Multidimensionality of the underlying process is also believed to be advantageous to daGOAT when sample size is limited. When most of the observed features independently contribute to relative risk, there is little benefit for a model to distinguish between true effectors and dummies. Therefore, daGOAT does not conduct any variable selection, while – despite the small sample size and the time-varying nature of feature contributions to relative risk – on each day XGBoost would have to pick a new set of key variables to grow trees.
2.06	Authors proposed method is validated with the performance of stationary features only model, landmark specific plasma biomarker levels and landmark-specific random survival forest models [Lines 115- 118]. There is no comparison of the proposed approach with the state-of-the-art non-parametric approaches. Why does the proposed method is not compared to other state-of-	We thank Reviewer #2 for calling this out. We were originally searching in the survival analysis literature for a benchmark algorithm, and what we found was Random Survival Forests (RSF) (Ishwaran et al. 2008). We are no longer doing survival analysis and now choose to benchmark against XGBoost, a state-of-the-art tree boosting algorithm. See Figs. 1 & 3.

	the-art non-parametric approaches? More experiments on the state-of-the-art approaches using aGOAT dataset and their comparison with the proposed approach will be better.	
3.01	This reviewer wonder how this technics can really be applied to the clinics, i.e. this predictive score has instantly impacts on the patient therapeutic procedures. At least the mechanism to calculate the score more easily is mandatory. Please discuss more on this point. This paper may aim to apply their methods to the actual clinical field (bedsides of the patient), and to be helpful in the clinical decision. However, their calculation methods using complicated computing systems seem to be too complicated. That is why the method to calculate the scores which they invented more easily looks necessary.	The following paragraph is now included in the revised manuscript to better explain how deployment of the daGOAT in real clinical settings might look like (lines 391–406): For deployment in clinical settings, the daGOAT model must be integrated into the hospital informatics system. All input data we utilized in this study are readily accessible through the electronic health records at the IHCAMS. On any given day, we have ≈ 100 patients who have just undergone HSCT (within 100 days) and are still hospitalized at the IHCAMS; these are the patients whose dynamic clinical data need to be updated daily. Our semi-automatic data process takes less than 30 minutes to extract the 100 patients' newly collected data on the latest day from the electronic health records and subsequently append the new incoming data to patients' multidimensional time-series data frame accumulated in previous days. The daGOAT model is fast to compute. On average, computing $\varphi_i(t)$'s for 100 consecutive days for one patient takes approximately half a second. The latest calculated risk scores can then be pushed to the healthcare team via a dialog box on the computer screen or via email alerts. Model-fitting is also reasonably fast. Fitting daGOAT on the adult training set ($n=584$, $p=194$, $T=100$), for instance, took less than one minute using a typical desktop computer. To sum up, daGOAT is easy to implement, provided that the hospital informatics system is

		sufficiently ‘modern’. We, too, prefer simpler models. At least in our dataset, however, both the two-biomarker MAGIC score and the three-biomarker Ann Arbor score failed to predict severe aGVHD (Fig. 1c). In addition, our experiments did not suggest much benefit in conducting variable selection in the context of severe aGVHD prediction (lines 197–204, lines 210–222, and Figs. 2a,b,d,e). Moreover, two new paragraphs are now included in Discussion to expand on the subject of model parsimony (lines 280–297).
3.02	As mentioned, this can be a good model as a methodology and their paper is successful as a proof-of-concept study. However, the authors focused on the very minute part of the biology (extremely rare); post-transplant complication for hematological malignancies. The Reviewer is not sure their methodology can be applied to more generalized topics (ie. the analysis of more common diseases, like hypertension, strokes, or some cancers), which make the reviewer suggest that this paper should be submitted to the other more specific journal (that covers hematology or transplantation) not to the general journal of Nature computation science.	We thank Reviewer #3 for prompting us to think hard on the important subject of generalizability. We are a hematological center, and pretty much all the diseases we study are ‘rare’ according to the definition (affecting less than 1/1500 of the population) stated in the Orphan Drug Act of 1983 passed by the U.S. Congress. Hematological datasets have just as many ‘-omics’ features as in other diseases that are more prevalent, but sample sizes in hematology are often smaller by 1 or 2 orders of magnitude. We are passionate about devising computational strategies that can help clinicians tackle such ‘large p, small n’ problems. We do not believe a hematology-centric journal would be a more suitable venue to publish our work; after all, our study is highly geared towards computation. Rather, we think it is important to elevate the computational community’s general awareness of the statistical challenges we are encountering in rare emergencies. That we have received insightful feedback during peer review at Nature Computational Science also attests the importance for us to reach out to the broader computational community. After some serious thinking, we realized: the main

		reason we were not persuasive in demonstrating the generalizability of daGOAT in our previous manuscript is that we ourselves did not understand when and why daGOAT worked. We have since conducted a series of simulation experiments (lines 224-246, lines 366-379, lines 505-546, and Figs. 3a) to interrogate daGOAT in various data scenarios and reached the conclusion that daGOAT is more suited for tackling a ‘large p, small n’ time-series problem when the underlying process for event generation is smooth and multidimensional and when many features are missing values at multiple time points. Fig. 1b and Figs. 2a,b suggested that the aGOAT dataset indeed falls under this category. In other words, we now have a few rules of thumb to assist in deciding when the daGOAT algorithm can be considered in solving forecasting problems.
--	--	---

Decision Letter, first revision:

Date: 7th February 22 13:29:29
Last Sent: 7th February 22 13:29:29
Triggered By: Kaitlin McCardle
From: kaitlin.mccardle@us.nature.com
To: chenjunren@ihcams.ac.cn
CC: computacionalscience@nature.com
BCC: kaitlin.mccardle@us.nature.com
Subject: AIP Decision on Manuscript NATCOMPUTSCI-21-0881A
Message: Our ref: NATCOMPUTSCI-21-0881A

7th February 2022

Dear Dr. CHEN,

Thank you for submitting your revised manuscript "Dynamic forecasting of severe acute graft-versus-host disease after transplantation" (NATCOMPUTSCI-21-0881A). It has now been seen by the original referees and their comments are below. The reviewers find that the paper has improved in revision, and therefore we'll be happy in principle to publish it in Nature Computational Science, pending minor revisions to satisfy the referees' final requests and to comply with our editorial and formatting guidelines. Please note that we are still corresponding with the reviewers to confirm that there are no additional requests regarding your code.

TRANSPARENT PEER REVIEW

Nature Computational Science offers a transparent peer review option for new original research manuscripts submitted from 17th February 2021. We encourage increased transparency in peer review by publishing the reviewer comments, author rebuttal letters and editorial decision letters if the authors agree. Such peer review material is made available as a supplementary peer review file. **Please state in the cover letter 'I wish to participate in transparent peer review' if you want to opt in, or 'I do not wish to participate in transparent peer review' if you don't.** Failure to state your preference will result in delays in accepting your manuscript for publication. Please note: we allow redactions to authors' rebuttal and reviewer comments in the interest of confidentiality. If you are concerned about the release of confidential data, please let us know specifically what information you would like to have removed. Please note that we cannot incorporate redactions for any other reasons. Reviewer names will be published in the peer review files if the reviewer signed the comments to authors, or if reviewers explicitly agree to release their name. For more information, please refer to our [FAQ page](https://www.nature.com/documents/nr-transparent-peer-review.pdf).

Thank you again for your interest in Nature Computational Science Please do not hesitate to contact me if you have any questions.

Sincerely,

Kaitlin McCardle
Editor
Nature Computational Science

ORCID

Reviewer #1 (Remarks to the Author):

All concerns have been adequately addressed.

Reviewer #2 (Remarks to the Author):

Thanks for addressing the review comments. It substantially improved the manuscript and can be accepted in its current format.

Reviewer #3 (Remarks to the Author):

The revised manuscript has well responded to the critics from this Reviewer.

Author Rebuttal, first revision:

Response to referees

#	Reviewer's comment	Response
1	All concerns have been adequately addressed.	N/A
2	Thanks for addressing the review comments. It substantially improved the manuscript and can be accepted in its current format.	N/A
3	Remarks to the Author: The revised manuscript has well responded to the critics from this Reviewer. I have checked through the code, and the source code has no concerns on reproducibility. As for usability, the source code to some extent requires the experienced staff and there are some points to be revised. (for example, developing web-based apps or so. but not required in the revision of this study. It could be much better if the authors (or others) develop a more user-friendly system (like web apps).)	We acknowledge that providing full algorithmic details does not immediately lead to broad application of our model. Building an app that connects to other hospitals' information systems to extract their patients' clinical data (with non-unified data schemas) for updating daGOAT scores daily, however, is not an easy task and will require collaborative discussions on data privacy, data fire walls, data schema differences, cross-border legal matters, etc. These data access and data integration issues cannot be solved by the authors alone. We will incorporate our model in the hospital information system at the IHCAMS and will publish the results of clinical testing when it is appropriate to do so; any learned lessons on how to make it easier for more hospitals to implement daGOAT will also be shared with the scientific community.

Final Decision Letter:**Date:** 14th February 22 15:02:49**Last Sent:** 14th February 22 15:02:49**Triggered By:** Kaitlin McCardle**From:** kaitlin.mccardle@us.nature.com**To:** chenjunren@ihcams.ac.cn**Subject:** Decision on Nature Computational Science manuscript NATCOMPUTSCI-21-0881B**Message:** Dear Mr CHEN,

We are pleased to inform you that your Brief Communication "Dynamic forecasting of severe acute graft-versus-host disease after transplantation" has now been accepted for publication in Nature Computational Science.

Please note that *Nature Computational Science* is a Transformative Journal (TJ). Authors may publish their research with us through the traditional subscription access route or make their paper immediately open access through payment of an article-processing charge (APC). Authors will not be required to make a final decision about access to their article until it has been accepted. [Find out more about Transformative Journals](https://www.springernature.com/gp/open-research/transformative-journals)

Authors may need to take specific actions to achieve [compliance with funder and institutional open access mandates](https://www.springernature.com/gp/open-research/funding/policy-compliance-faqs). For submissions from January 2021, if your research is supported by a funder that requires immediate open access (e.g. according to [Plan S principles](https://www.springernature.com/gp/open-research/plan-s-compliance)) then you should select the gold OA route, and we will direct you to the compliant route where possible. For authors selecting the subscription publication route our standard licensing terms will need to be accepted, including our [self-archiving policies](https://www.springernature.com/gp/open-research/policies/journal-policies). Those standard licensing terms will supersede any other terms that the author or any third party may assert apply to any version of the manuscript.

Acceptance of your manuscript is conditional on all authors' agreement with our

publication policies (see <https://www.nature.com/natcomputsci/for-authors>). In particular your manuscript must not be published elsewhere and there must be no announcement of the work to any media outlet until the publication date (the day on which it is uploaded onto our web site).

Before your manuscript is typeset, we will edit the text to ensure it is intelligible to our wide readership and conforms to house style. We look particularly carefully at the titles of all papers to ensure that they are relatively brief and understandable.

Once your manuscript is typeset and you have completed the appropriate grant of rights, you will receive a link to your electronic proof via email with a request to make any corrections within 48 hours. If, when you receive your proof, you cannot meet this deadline, please inform us at rjsproduction@springernature.com immediately.

If you have queries at any point during the production process then please contact the production team at rjsproduction@springernature.com. Once your paper has been scheduled for online publication, the Nature press office will be in touch to confirm the details.

Content is published online weekly on Mondays and Thursdays, and the embargo is set at 16:00 London time (GMT)/11:00 am US Eastern time (EST) on the day of publication. If you need to know the exact publication date or when the news embargo will be lifted, please contact our press office after you have submitted your proof corrections. Now is the time to inform your Public Relations or Press Office about your paper, as they might be interested in promoting its publication. This will allow them time to prepare an accurate and satisfactory press release. Include your manuscript tracking number NATCOMPUTSCI-21-0881B and the name of the journal, which they will need when they contact our office.

About one week before your paper is published online, we shall be distributing a press release to news organizations worldwide, which may include details of your work. We are happy for your institution or funding agency to prepare its own press release, but it must mention the embargo date and Nature Computational Science. Our Press Office will contact you closer to the time of publication, but if you or your Press Office have any inquiries in the meantime, please contact press@nature.com.

We welcome the submission of potential cover material (including a short caption of around 40 words) related to your manuscript; suggestions should be sent to Nature Computational Science as electronic files (the image should be 300 dpi at 210 x 297 mm in either TIFF or JPEG format). We also welcome suggestions for the Hero Image, which appears at the top of our [home page](http://www.nature.com/natcomputsci); these should be 72 dpi at 1400 x 400 pixels in JPEG format. Please note that such pictures should be selected more for their aesthetic appeal than for their scientific content, and that colour images work better than black and white or grayscale images. Please do not try

to design a cover with the Nature Computational Science logo etc., and please do not submit composites of images related to your work. I am sure you will understand that we cannot make any promise as to whether any of your suggestions might be selected for the cover of the journal.

Best regards,

Kaitlin McCardle
Editor
Nature Computational Science

P.S. Click on the following link if you would like to recommend Nature Computational Science to your librarian: https://www.springernature.com/gp/librarians/recommend-to-your-library

** Visit the Springer Nature Editorial and Publishing website at www.springernature.com/editorial-and-publishing-jobs for more information about our career opportunities. If you have any questions please click here. **